# ENERGY-WEIGHTED FLOW MATCHING: UNLOCKING CONTINUOUS NORMALIZING FLOWS FOR EFFICIENT AND SCALABLE BOLTZMANN SAMPLING

## ABSTRACT

Sampling from unnormalized target distributions, e.g. Boltzmann distributions $\mu_{\text{target}}(x) \propto \exp(-E(x)/T)$, is fundamental to many scientific applications yet computationally challenging due to complex, high-dimensional energy landscapes. Existing approaches applying modern generative models to Boltzmann distributions either require large datasets of samples drawn from the target distribution or, when using only energy evaluations for training, cannot efficiently leverage the expressivity of advanced architectures like continuous normalizing flows that have shown promise for molecular sampling. To address these shortcomings, we introduce Energy-Weighted Flow Matching (EWFM), a novel training objective enabling continuous normalizing flows to model Boltzmann distributions using only energy function evaluations. Our objective reformulates conditional flow matching via importance sampling, allowing training with samples from arbitrary proposal distributions. Based on this objective, we develop two algorithms: iterative EWFM (iEWFM), which progressively refines proposals through iterative training, and annealed EWFM (aEWFM), which additionally incorporates temperature annealing for challenging energy landscapes. On benchmark systems, including challenging 55-particle Lennard-Jones clusters, our algorithms demonstrate sample quality competitive with state-of-the-art energy-only methods while requiring up to three orders of magnitude fewer energy evaluations.

## 1 INTRODUCTION

Understanding the behavior of systems with many interacting particles is a central task in many scientific fields, ranging from molecular dynamics (Allen & Tildesley, 2017; Frenkel & Smit, 2023) to computational chemistry (Shirts & Chodera, 2008; Noé et al., 2019) and protein science (Bryngelson et al., 1995; Dill et al., 2008). In these multi-particle systems, the equilibrium distribution of configurations $x$ (e.g., positions of atoms in a molecule) is often governed by a known energy function $E(x)$, giving rise to a Boltzmann distribution with unnormalized density $\mu_{\text{target}}(x) \propto \exp(-E(x)/T)$, where $T$ is the system's temperature. Generating independent samples from this distribution is essential for computing equilibrium properties, such as the probability of a protein being in a folded state. However, this task remains computationally challenging for complex, high-dimensional systems.

Traditional trajectory-based methods such as Markov Chain Monte Carlo (MCMC) (Hastings, 1970; Andrieu et al., 2003) and molecular dynamics (MD) (Leimkuhler & Matthews, 2013) address this challenge by simulating paths through the system's energy landscape. However, these energy landscapes exhibit numerous local minima (metastable states) separated by high-energy barriers, often causing simulated trajectories to remain trapped within local minima for long periods. This typically leads to prohibitively long simulation times to adequately explore the entire distribution (Allen & Tildesley, 2017; Noé et al., 2019; Łatuszyński et al., 2025; Pompe et al., 2020).

While deep generative models (Lipman et al., 2024; Papamakarios et al., 2021; Yang et al., 2023) offer a modern alternative for learning and sampling from complex distributions, they cannot be directly applied to Boltzmann distributions as they require training samples from the target distribution — precisely what we seek to generate in the first place. To address this circular problem, Noé et al.

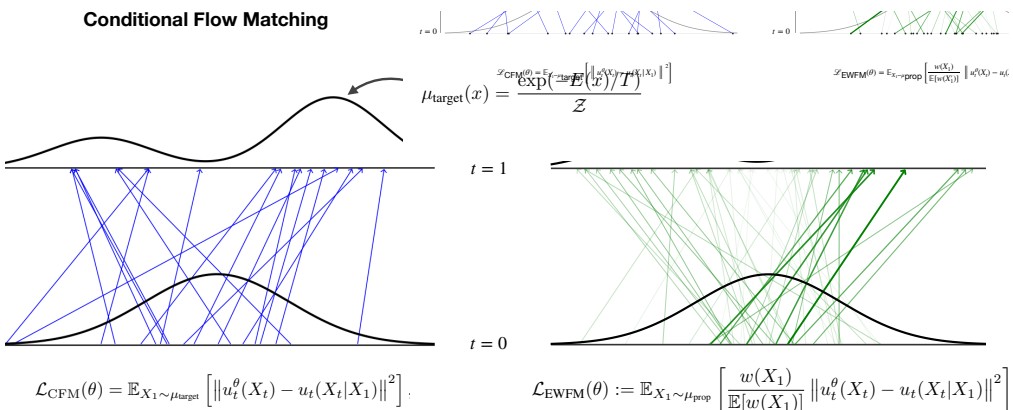

Figure 1: **Conditional Flow Matching vs. Energy-Weighted Flow Matching.** *(Left)* Conditional Flow Matching (CFM) requires samples from the target distribution $\mu_{\text{target}}$. The model learns by regressing on points $x_t$ along conditional paths from prior $p_0$ to target samples. *(Right)* Energy-Weighted Flow Matching (EWFM) reformulates the CFM objective to avoid requiring target samples, instead using an arbitrary proposal distribution. Training points are reweighted by importance weights $w(x_1)$ of their endpoints. High-weight paths (thick lines) are amplified while low-weight paths (thin lines) are suppressed, yielding an equivalent objective that learns the target distribution.

(2019) introduced *Boltzmann generators*, a class of deep generative models based on normalizing flows (Rezende & Mohamed, 2015; Dinh et al., 2016) that primarily leverage the known energy function for training via minimizing the reverse KL divergence between the model and the unnormalized target Boltzmann density. Nevertheless, these methods still require some initial target data to supplement the energy-based training, as the reverse KL divergence alone leads to incomplete target coverage due to its mode-seeking behavior.

This has motivated a line of research into methods that train using only energy function evaluations, without requiring any target samples. Within this line of research, the current state-of-the-art is represented by Flow Annealed Importance Sampling Bootstrap (FAB) (Midgley et al., 2022) and Iterated Denoising Energy Matching (iDEM) (Akhound-Sadegh et al., 2024).[1] FAB combines normalizing flows with annealed importance sampling, achieving efficiency advantages over many prior methods, but still faces scalability challenges for very high-dimensional systems. iDEM trains diffusion models by regressing against Monte Carlo estimators of the score function, demonstrating high-quality sampling on large-dimensional systems, though the Monte Carlo estimation requires substantial energy evaluations during training.

In parallel, researchers have explored leveraging more advanced and expressive generative modeling approaches for Boltzmann distributions, particularly Continuous Normalizing Flows (CNFs) (Chen et al., 2018; Cornish et al., 2020) trained via Flow Matching (Lipman et al., 2022; Albergo et al., 2023). These approaches have achieved notable successes, including the first direct sampling of molecular equilibrium distributions in Cartesian coordinates (Klein et al., 2023) and the development of transferable models that can generalize across diverse chemical systems (Klein & Noé, 2024; Vaitl & Klein, 2025). However, despite their considerable promise for Boltzmann sampling, current flow matching formulations still require large datasets of samples from the target distribution for training. This creates a fundamental challenge for complex Boltzmann distributions, as we ideally need both the expressivity of advanced architectures and the capability of efficient energy-only training.

**Contributions.** To overcome the reliance on target data for training continuous normalizing flows for Boltzmann sampling, we introduce the Energy-Weighted Flow Matching (EWFM) framework, establishing a new approach for scalable and efficient energy-only Boltzmann generators. Concretely, we make the following contributions:

---

[1]The field is currently moving very fast, with multiple concurrent works recently improving upon the previous state-of-the-art (see concurrent work paragraph in Sec. 5).

First, we introduce the *Energy-Weighted Flow Matching (EWFM) objective*, $\mathcal{L}_{\text{EWFM}}$ (Sec. 3.1). This objective reformulates the standard flow matching loss to be independent of target samples by utilizing data from an arbitrary proposal distribution, $\mu_{\text{prop}}$. Inspired by similar energy-based reweighting strategies (Zhang et al., 2025), our objective then corrects for the mismatch by reweighting the loss for each sample $x$ based on its Boltzmann weight, $\exp(-E(x)/T)$, and its proposal density, $\mu_{\text{prop}}(x)$ (see Fig. 1 for a visual comparison with standard conditional flow matching).[2]

Based on this objective, we introduce two algorithms. The *iterative EWFM (iEWFM) algorithm* (Sec. 3.2) employs an iterative refinement strategy where the model trained in one step serves as a better-informed proposal for the next, allowing progressive learning of the target distribution. As a principled extension for challenging energy landscapes, the *annealed EWFM (aEWFM) algorithm* (Sec. 3.3) incorporates temperature annealing, i.e., first training on a smoothed energy landscape at high temperature $T_{\text{init}} > T$ before gradually cooling to the target temperature.

Finally, in Sec. 4, we provide empirical validation demonstrating competitive performance on benchmark Boltzmann distributions, including Gaussian mixtures and n-body systems (4-particle double-well, 13- and 55-particle Lennard-Jones). Our algorithms achieve sample quality comparable to state-of-the-art methods while requiring significantly fewer energy evaluations than iDEM and demonstrating better scalability than FAB.

## 2 BACKGROUND AND PRELIMINARIES

We aim to generate i.i.d. samples from a Boltzmann distribution $\mu_{\text{target}}$ over $\mathbb{R}^d$ defined as:

$$\mu_{\text{target}}(x) = \frac{\exp(-E(x)/T)}{\mathcal{Z}}, \quad \mathcal{Z} = \int_{\mathbb{R}^d} \exp(-E(x)/T)\, dx. \tag{1}$$

Here, $E(x) : \mathbb{R}^d \to \mathbb{R}$ is the energy function of the system, and $T$ is the temperature. The denominator $\mathcal{Z}$ is the partition function, which is generally intractable to compute for high-dimensional systems. Instead, we have access to the energy function $E(x)$ for any configuration $x$, allowing us to evaluate the unnormalized density $\exp(-E(x)/T)$.

### 2.1 BOLTZMANN GENERATORS

Boltzmann generators are a class of generative models trained to produce independent samples from the Boltzmann distribution. By learning a direct transformation from a simple prior to the target distribution, they generate samples in a single pass, contrasting with trajectory-based simulations requiring many sequential steps.

The core contribution of Boltzmann generators, introduced by Noé et al. (2019), is to adapt generative model training (in this case, normalizing flows) to settings where only the energy function $E(x)$ and temperature $T$ are known. This approach minimizes the reverse KL divergence, $\text{KL}(q_\theta \| \mu_{\text{target}})$, which can be estimated using samples from the model $q_\theta$ itself

$$\text{KL}(q_\theta \| \mu_{\text{target}}) = \mathbb{E}_{X \sim q_\theta} \left[ \log q_\theta(X) + E(X)/T \right] + \log \mathcal{Z}. \tag{2}$$

Since $\log \mathcal{Z}$ is constant with respect to model parameters $\theta$, it can be ignored during optimization. To stabilize training and address the mode-seeking behavior of reverse KL, this method combines forward and reverse KL divergences using a small set of initial target samples.

A key application of trained Boltzmann generators $q_\theta$ is computing equilibrium properties using self-normalized importance sampling (SNIS), where model samples are reweighted to obtain asymptotically unbiased estimates (Nicoli et al., 2020; Noé et al., 2019). The importance weights are $w(x) = \exp(-E(x)/T)/q_\theta(x)$, and observables are estimated as $\mathbb{E}_{x \sim \mu_{\text{target}}}[O(x)] \approx \frac{\sum_{i=1}^{N} w(x_i)O(x_i)}{\sum_{i=1}^{N} w(x_i)}$ where $x_i \sim q_\theta$.

### 2.2 CONTINUOUS NORMALIZING FLOWS AND FLOW MATCHING

Continuous Normalizing Flows (CNFs) (Chen et al., 2018) extend classical normalizing flows by defining transformations as solutions to ordinary differential equations (ODEs). They offer greater

---

[2]We use $\mu_{\text{prop}}$ interchangeably for the proposal distribution and its density throughout the paper, and similarly for other distributions.

flexibility than traditional normalizing flows by relaxing strict invertibility constraints and enabling more expressive neural network architectures. More concretely, CNFs learn a time-dependent vector field $u_t^\theta(x)$, parameterized by a neural network with parameters $\theta$. This vector field induces a continuous transformation, or flow, $\psi : [0, 1] \times \mathbb{R}^d \to \mathbb{R}^d$ that maps a simple base distribution $p_0$ to a complex target distribution $p_1$. The transformation is governed by the ODE

$$\frac{d}{dt}\psi_t(x) = u_t^\theta(\psi_t(x)), \quad \text{with initial condition} \quad \psi_0(x) = x. \tag{3}$$

The flow induces a *probability path* $(p_t)_{t \in [0,1]}$, where $p_t$ is the probability distribution of $\psi_t(X_0)$ for $X_0 \sim p_0$, providing a continuous sequence of distributions that interpolates between $p_0$ and $p_1$. Given a learned vector field $u_t^\theta$, the Instantaneous Change of Variables Formula Chen et al. (2018) allows for exact likelihood computation at any point $\psi_1(x)$ in the target distribution[3] via

$$\log p_1(\psi_1(x)) = \log p_0(x) - \int_0^1 \text{div}(u_t^\theta)(\psi_t(x)) \, dt. \tag{4}$$

To avoid computationally expensive ODE solving for likelihood computation during training, the flow matching paradigm (Lipman et al., 2022; Albergo et al., 2023) offers an efficient alternative. The core idea is to regress the parameterized vector field $u_t^\theta$ onto a target vector field $u_t$ that generates a desired probability path between $p_0$ and $p_1$. Although the ideal vector field is generally intractable to compute, this problem is circumvented by instead optimizing the equivalent *conditional* flow matching objective

$$\mathcal{L}_{\text{CFM}}(\theta) = \mathbb{E}_{t,X_t,X_1}\left[\left\|u_t^\theta(X_t) - u_t(X_t|X_1)\right\|^2\right], \text{ where } t \sim U[0,1], X_1 \sim p_1, X_t \sim p_{t|1}(\cdot|X_1). \tag{5}$$

This reduces flow matching to a regression task, enabling simulation-free training. However, this approach requires target samples $X_1$ — precisely what we lack in Boltzmann sampling.

## 3 THE ENERGY-WEIGHTED FLOW MATCHING FRAMEWORK

We now introduce the Energy-Weighted Flow Matching framework, which includes the Energy-Weighted Flow Matching (EWFM) objective that enables CNF training without target samples, and two algorithms that leverage this objective: iterative EWFM and annealed EWFM.

### 3.1 THE ENERGY-WEIGHTED FLOW MATCHING OBJECTIVE

As established in Sec. 2.2, the standard conditional flow matching loss requires access to samples from the target distribution $\mu_{\text{target}}$. In the context of Boltzmann sampling, however, such samples are unavailable. To bridge this gap, we reformulate the CFM loss into an equivalent objective where the expectation over the intractable target distribution is replaced by an expectation over an arbitrary proposal distribution $\mu_{\text{prop}}$. We can interpret this from the perspective of importance sampling.

The key insight is that while we cannot sample from the target distribution, we can evaluate its unnormalized density $\exp(-E(x)/T)$. This enables us to rewrite the CFM loss as an expectation over an arbitrary proposal distribution using importance weights. More formally, the conditional flow matching loss can be decomposed as $\mathcal{L}_{\text{CFM}}(\theta) = \mathbb{E}_{X_1 \sim \mu_{\text{target}}}[f(X_1; \theta)]$, where $f(x_1; \theta)$ represents the expected loss conditioned on endpoint $x_1$. By using the relationship

$$\frac{\mu_{\text{target}}(x_1)}{\mu_{\text{prop}}(x_1)} = \frac{w(x_1)}{\mathbb{E}_{X_1' \sim \mu_{\text{prop}}}[w(X_1')]} \tag{6}$$

with unnormalized weights $w(x_1) = \frac{\exp(-E(x_1)/T)}{\mu_{\text{prop}}(x_1)}$, we can take the expectation in the CFM loss with respect to an arbitrary proposal distribution to obtain our Energy-Weighted Flow Matching (EWFM) objective:

$$\mathcal{L}_{\text{EWFM}}(\theta; \mu_{\text{prop}}) := \mathbb{E}_{t,X_t,X_1}\left[\frac{w(X_1)}{\mathbb{E}_{X_1' \sim \mu_{\text{prop}}}[w(X_1')]}\left\|u_t^\theta(X_t) - u_t(X_t|X_1)\right\|^2\right] \tag{7}$$

$$= \mathbb{E}_{X_1 \sim \mu_{\text{prop}}}\left[\frac{w(X_1)}{\mathbb{E}_{X_1' \sim \mu_{\text{prop}}}[w(X_1')]}f(X_1; \theta)\right] = \mathcal{L}_{\text{CFM}}(\theta).$$

---

[3]Here, $\psi_1(x)$ is the final point after solving the ODE forward from initial point $x$.

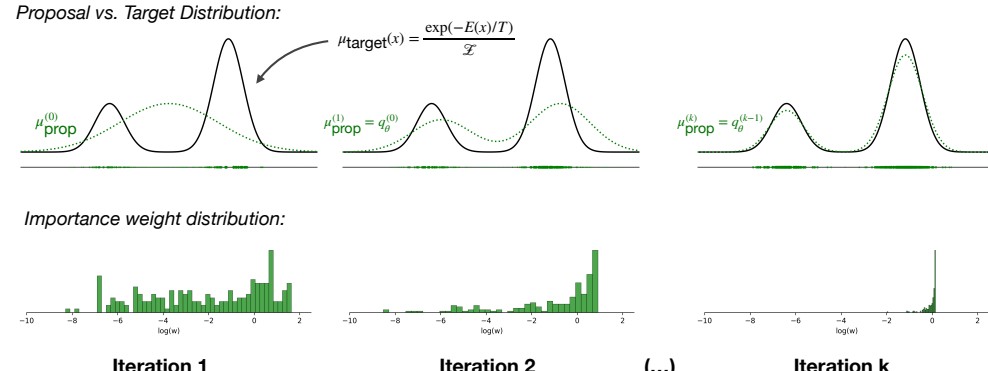

Figure 2: **The iterative EWFM algorithm.** *(Top row)* Shows target distribution $\mu_{\text{target}}$ (solid black) and proposal distribution $\mu_{\text{prop}}$ (dotted green) for each iteration, with samples displayed as dots whose size reflects their importance weights. *(Bottom row)* Corresponding distribution of log importance weights. *Iteration 1:* Initial proposal (single Gaussian) poorly matches the target, resulting in highly variable importance weights. *Iteration 2:* Using the previous model as proposal shows improvement — better capturing target modes with more balanced weights. *Iteration k:* After convergence, the proposal closely matches the target, yielding low-variance weights and stable training.

Here, we assume that $t \sim U[0,1]$, $X_1 \sim \mu_{\text{prop}}$, and $X_t \sim p_{t|1}(\cdot|X_1)$. The mathematical equivalence $\mathcal{L}_{\text{CFM}}(\theta) = \mathcal{L}_{\text{EWFM}}(\theta; \mu_{\text{prop}})$ ensures that minimization of our importance-weighted objective has the same theoretical minimum as the original CFM loss. The derivation is provided in Appx. B.1.

Fig. 1 illustrates the differences between standard CFM and our EWFM approach. A similar reweighting strategy was used by Zhang et al. (2025) to steer a learned base distribution towards a target proportional to $p_1(x) \exp(-E(x))$ for reinforcement learning applications.

**Remark.** Crucially, this differs from methods that regress to the marginal score/vector field, such as iDEM (Akhound-Sadegh et al., 2024) or the recently proposed Iterated Energy-based Flow Matching (iEFM) (Woo & Ahn, 2024), which require a nested Monte Carlo importance sampling loop to approximate the marginal score/vector field. Instead, EWFM applies importance sampling directly to the objective. By regressing to the exact conditional vector field, we eliminate the inner estimation loop, reducing the cost to a single energy evaluation per trajectory.

### 3.2 THE ITERATIVE ENERGY-WEIGHTED FLOW MATCHING ALGORITHM

While the EWFM objective is theoretically sound, its practical estimation via Monte Carlo methods introduces a challenge. If the proposal distribution $\mu_{\text{prop}}$ differs substantially from the target $\mu_{\text{target}}$, the importance weights $w(x)$ will have high variance. This means the Monte Carlo estimate will be dominated by a few samples with extremely large weights, leading to unstable gradients and ineffective training.

To address this, we introduce the *iterative EWFM (iEWFM) algorithm*. The core idea is to use the current generative model $q_\theta$ as the proposal distribution for the next training step. We start with an initial proposal distribution (e.g., a simple Gaussian), train a model using EWFM, and then use this trained model as the proposal for the next iteration. Each iteration produces a better approximation to the target distribution, which serves as a higher-quality proposal for subsequent training.

We motivate this iterative strategy through importance sampling theory. The Monte-Carlo estimate of the gradient of the EWFM objective can be written as:

$$\hat{\nabla}_\theta \mathcal{L}_{\text{EWFM}} = \sum_{n=1}^{N} \tilde{w}^{(n)} \phi_\theta(x^{(n)}), \quad \text{where} \quad \tilde{w}^{(n)} = \frac{w(x^{(n)})}{\sum_{m=1}^{N} w(x^{(m)})}. \tag{8}$$

where $\phi_\theta(x_1)$ represents the gradient of the loss conditioned on endpoint $x_1$. From importance sampling theory, the optimal proposal distribution that minimizes the variance of such estimators

**Algorithm 1:** Iterative Energy-Weighted Flow Matching (iEWFM) - Simplified

---

**Input:** Energy function $E(x)$, temperature $T$, initial proposal $\mu_{\text{prop}}^{(0)}$
**Output:** Trained model $q_\theta$

$\mu_{\text{prop}} \leftarrow \mu_{\text{prop}}^{(0)}$; Generate initial buffer $\mathcal{B}$ from $\mu_{\text{prop}}$;
**for** *each epoch* **do**
    **if** *time to refresh buffer* **then**
        $\mu_{\text{prop}} \leftarrow q_\theta$ (use current model as proposal);
        Generate buffer $\mathcal{B}$ from $\mu_{\text{prop}}$;
    **end**
    Sample mini-batch $\{x_1^{(n)}\}$ from buffer $\mathcal{B}$;
    Compute importance weights $w^{(n)} = \exp(-E(x_1^{(n)})/T - \log \mu_{\text{prop}}(x_1^{(n)}))$;
    Compute CFM gradients $\hat{\phi}_\theta(x_1^{(n)})$ for each $x_1^{(n)}$;
    Update $\theta$ using SNIS gradient $\hat{\nabla}_\theta \mathcal{L}_{\text{EWFM}} = \frac{\sum_n \hat{\phi}_\theta(x_1^{(n)}) w^{(n)}}{\sum_m w^{(m)}}$;
**end**

---

is given by $\mu_{\text{opt}}(x) \propto \mu_{\text{target}}(x) \cdot \|\phi_\theta(x) - \nabla_\theta \mathcal{L}_{\text{EWFM}}\|$. Under the simplifying assumption that the term $\|\phi_\theta(x) - \nabla_\theta \mathcal{L}_{\text{EWFM}}\|$ does not vary substantially across the domain, this reduces to being approximately proportional to the target density (see Appx. B.2 for the complete derivation).

Since our model $q_\theta$ is trained to approximate $\mu_{\text{target}}$, this motivates our iterative strategy of using the current model as the proposal for the next training step. As training progresses, $q_\theta$ becomes a progressively better approximation of the target, leading to lower-variance gradient estimates and more stable training. Fig. 2 illustrates this iterative refinement process. Note that our approach continually refines a single model throughout the iterative training process, rather than training separate models at each iteration.

**Amortized training with a sample buffer.** Since evaluating the proposal density $q_\theta(x)$ requires solving the reverse-time ODE of the CNF and computing the log-likelihood via Eq. (4), which is computationally expensive, we amortize these costs using a sample buffer. We periodically generate samples, pre-compute their log-densities and energies, and reuse them across multiple training steps. This buffering strategy, which has been similarly employed in related work (Midgley et al., 2022; Akhound-Sadegh et al., 2024), preserves the adaptive proposal benefits while improving efficiency. The complete iEWFM algorithm incorporating this buffering approach is presented in Alg. 1, with full implementation details provided in Alg. 2 in the appendix.

### 3.3 ANNEALED EWFM: SCALING TO COMPLEX ENERGY LANDSCAPES

While iEWFM provides a robust strategy for the Boltzmann distributions investigated in this work, the quality of the initial proposal can become a limiting factor for more challenging energy landscapes. For such systems, a randomly initialized model forms a poor initial proposal, leading to high-variance gradient estimates that prevent the iterative algorithm from effectively improving. To overcome this bootstrapping problem, we extend iEWFM with temperature annealing. Rather than forcing a randomly initialized proposal to fit a difficult target, we first train at an elevated temperature $T_0 > T$. At this higher temperature, the target distribution $\mu_{T_0}(x) \propto \exp(-E(x)/T_0)$ has flatter energy wells, increasing the likelihood that even a randomly initialized proposal achieves non-negligible overlap with the target, yielding lower-variance importance weights and more stable gradient estimates.

This insight leads to the *annealed Energy-Weighted Flow Matching (aEWFM)* algorithm. We employ a decreasing temperature schedule $T_0 > T_1 > \cdots > T_K = T$, implemented as a geometric progression. The aEWFM algorithm applies an equivalent iterative scheme to iEWFM by using the model from the previous step (either the previous temperature or the previous iteration at the same temperature) as the proposal.

324
325
326
327
328
329
330
331
332
333
334
335
336
337
338
339
340
341
342
343
344
345
346
347
348
349
350
351
352
353
354
355
356
357
358
359
360
361
362
363
364
365
366
367
368
369
370
371
372
373
374
375
376
377

Table 1: **Quantitative comparison with state-of-the-art Boltzmann sampling methods.** Results are reported as mean $\pm$ standard deviation over three random seeds. Bold indicates results not significantly outperformed by any other method in the same column ($p < 0.10$, Welch's t-test).

| Method | GMM-40 ($d = 2$) | | DW-4 ($d = 8$) | | LJ-13 ($d = 39$) | | LJ-55 ($d = 165$) | |
|---|---|---|---|---|---|---|---|---|
| | NLL $\downarrow$ | $\mathcal{W}_2 \downarrow$ | NLL $\downarrow$ | $\mathcal{W}_2 \downarrow$ | NLL $\downarrow$ | $\mathcal{W}_2 \downarrow$ | NLL $\downarrow$ | $\mathcal{W}_2 \downarrow$ |
| FAB | 7.14 ±0.01 | 12.00 ±5.73 | **7.16** ±0.01 | **2.15** ±0.02 | **17.52** ±0.17 | 4.35 ±0.01 | 200.32 ±62.30 | 18.03 ±1.21 |
| iDEM | **6.96** ±0.07 | 7.42 ±3.44 | **7.17** ±0.00 | **2.13** ±0.04 | **17.68** ±0.14 | 4.26 ±0.03 | 125.86 ±18.03 | **16.13** ±0.07 |
| EWFM (Ours) | 7.05 ±0.05 | **3.88** ±0.53 | 7.76 ±0.09 | **2.13** ±0.00 | 54.19 ±6.05 | 7.39 ±0.04 | - | - |
| iEWFM (Ours) | 7.08 ±0.03 | 6.68 ±0.67 | 7.65 ±0.13 | 2.25 ±0.03 | 19.38 ±1.27 | **4.19** ±0.06 | **97.66** ±2.38 | 16.38 ±0.13 |
| aEWFM (Ours) | 7.09 ±0.02 | 7.06 ±0.49 | 7.81 ±0.17 | 2.27 ±0.03 | 19.41 ±1.11 | **4.25** ±0.04 | 100.89 ±1.46 | **16.13** ±0.07 |

# 4 EXPERIMENTAL RESULTS

This section provides an empirical evaluation of our Energy-Weighted Flow Matching framework. We benchmark iEWFM and aEWFM against state-of-the-art methods for Boltzmann sampling without target data, analyzing sample quality, computational efficiency, and the contributions of our algorithmic components.

## 4.1 EXPERIMENTAL SETUP

We evaluate on four benchmarks: GMM-40 (2D, 40 components), DW-4 (4-particle double-well, 8D), LJ-13 (13-particle Lennard-Jones, 39D), and LJ-55 (55-particle Lennard-Jones, 165D). The Lennard-Jones systems are particularly challenging due to high dimensionality and sharp, multi-modal landscapes. We compare against two state-of-the-art energy-only methods: Flow Annealed Importance Sampling Bootstrap (FAB) (Midgley et al., 2022) and Iterated Denoising Energy Matching (iDEM) (Akhound-Sadegh et al., 2024), as well as EWFM, a simplified variant without iterative refinement, only using a simple baseline proposal. Sample quality is assessed using the 2-Wasserstein Distance ($\mathcal{W}_2$) and Negative Log-Likelihood (NLL), with energy function evaluations as our computational efficiency metric. For iDEM and FAB, we use the quantitative results reported in Akhound-Sadegh et al. (2024).

Following Akhound-Sadegh et al. (2024), we parameterize the vector field $u_t^\theta$ using MLPs with sinusoidal embeddings for GMM-40 and Equivariant Graph Neural Networks (Satorras et al., 2021) for particle systems. For aEWFM, we employ weight clipping and a geometric temperature annealing schedule. Complete implementation details and hyperparameters are provided in Appx. C.1 to C.3.

## 4.2 RESULTS

Table 1 presents our main quantitative results. Our methods achieve competitive performance across benchmarks, with particular strengths on complex systems. On GMM-40, EWFM excels on Wasserstein distance while remaining comparable on NLL, effectively capturing the mixture structure. Performance on DW-4 reveals limitations of our iterative approaches on this intermediate-complexity system, where baselines achieve stronger results. We hypothesize this may be due to potential bias in gradient estimates when using model proposals, though further investigation is needed to confirm this.

For high-dimensional Lennard-Jones systems, our iterative methods demonstrate clear advantages: both iEWFM and aEWFM achieve competitive or superior performance, with particularly strong results on LJ-55, where our methods substantially outperform iDEM (the previous state-of-the-art) on NLL, with aEWFM also matching it on Wasserstein distance. This confirms the effectiveness of our approach on challenging high-dimensional tasks.

The comparison between our variants reveals the importance of the iterative proposal scheme. While EWFM matches or outperforms iEWFM on simple systems (likely benefiting from exact density evaluation of the simple baseline proposal), it fails to converge on LJ-13, demonstrating that iterative refinement becomes essential for complex energy landscapes. Both iEWFM and aEWFM perform similarly across most systems, with aEWFM's robust performance indicating potential for

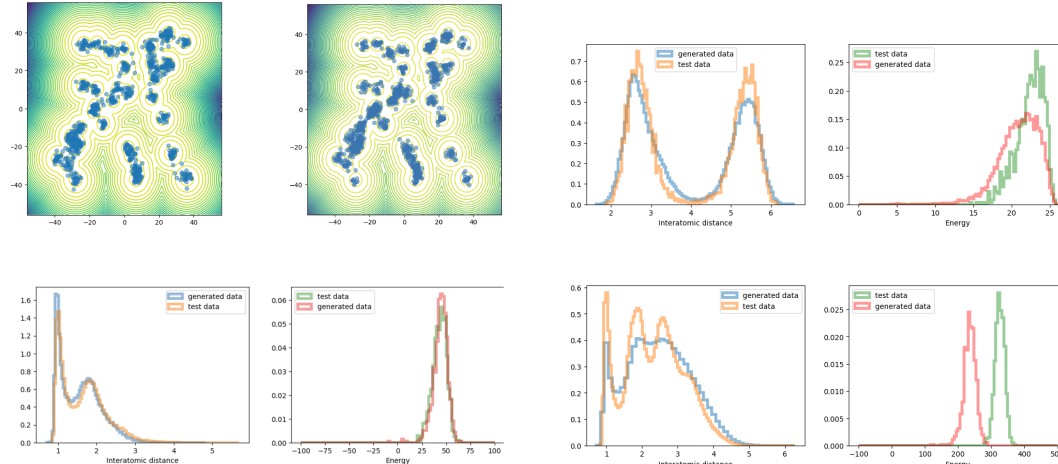

Figure 3: **Sample quality visualization across benchmark systems.** *(Top left)* EWFM samples and *(Top middle)* aEWFM samples for GMM-40, relatively accurately capturing all mixture components. *(Top right)* iEWFM performance on DW-4 showing distributions of interatomic distances and energy values, with limitations in capturing the correct relative weights between peaks. *(Bottom left)* aEWFM on LJ-13 shows excellent agreement with target distributions in both interatomic distance and energy distributions. *(Bottom right)* aEWFM performance on the challenging 165-dimensional LJ-55 system demonstrates relatively good performance despite the system's complexity.

our method to scale to even larger systems where iEWFM may struggle due to poor initial proposal quality.

**Energy Evaluation Efficiency.** A key advantage of our method is its energy evaluation efficiency, which is particularly important in real-world applications where energy function evaluations can be computationally expensive. As shown in Table 2, our methods require up to three orders of magnitude fewer energy evaluations than iDEM across all systems. For example, on LJ-13, iDEM requires $5 \times 10^{10}$ evaluations while our methods need only $1 \times 10^7$ —

Table 2: **Energy evaluations required during training.** EWFM variants require fewer energy evaluations than iDEM while being comparable to FAB across benchmark systems.

| Method | GMM | DW-4 | LJ-13 | LJ-55 |
|---|---|---|---|---|
| FAB | $1 \times 10^7$ | $1 \times 10^8$ | $6 \times 10^8$ | $6 \times 10^6$ |
| iDEM | $3 \times 10^{10}$ | $5 \times 10^{10}$ | $5 \times 10^{10}$ | $1 \times 10^9$ |
| EWFM variants | $3 \times 10^7$ | $3 \times 10^7$ | $1 \times 10^7$ | $1 \times 10^7$ |

a 5000-fold reduction. This efficiency likely stems from the EWFM objective providing more informative learning signals per evaluation than Monte-Carlo based objectives, as in iDEM, and our buffer-based strategy reusing samples across training steps. However, our approach comes with the trade-off of computationally intensive CNF density calculations, leading to longer wall-clock training times, though this trade-off becomes increasingly favorable for systems with computationally expensive energy functions.

## 5 RELATED WORK

Boltzmann generators (Noé et al., 2019) are deep generative models designed to overcome the computational expense of classical sampling methods for equilibrium distributions. Modern approaches can be broadly categorized by their data requirements: methods requiring target samples versus those using only energy evaluations. Among methods requiring target samples, we focus on flow matching approaches, which have recently shown particular promise for Boltzmann sampling.

**Flow Matching-Based Methods Requiring Target Data.** Recent advances in continuous normalizing flows trained with flow matching (Lipman et al., 2022; Albergo et al., 2023) for sampling from Boltzmann distributions have shown promising results. Klein et al. (2023) first sampled molecular

equilibrium distributions in Cartesian coordinates using equivariant flow matching with graph neural networks. This was extended to transferable Boltzmann generators that generalize across chemical space without retraining (Klein & Noé, 2024). Furthermore, Vaitl & Klein (2025) demonstrated that fine-tuning such models with path gradients can significantly improve sampling efficiency. Relatedly, Yu et al. (2024) proposed Force-guided Bridge Matching, a conditional bridge-matching framework that employs a hybrid approach combinig data and energy evaluations in a two stage training procedure. However, all these methods require large datasets of target samples, which is often not feasible in practice.

**Methods Using Only Energy Evaluations.** Several approaches train generative models using only energy function evaluations, with Flow Annealed Importance Sampling Bootstrap (FAB) (Midgley et al., 2022) and Iterated Denoising Energy Matching (iDEM) (Akhound-Sadegh et al., 2024) representing the current state-of-the-art in terms of sample quality and scalability. FAB combines normalizing flows with annealed importance sampling to address mode-seeking behavior but faces scalability challenges for very high-dimensional systems, while iDEM trains diffusion models using denoising energy matching with Monte Carlo score estimators, demonstrating scalability to large-dimensional systems though requiring substantial energy evaluations during training. In the same spirit, Noised Energy Matching and its bootstrapped variant BNEM (OuYang et al., 2025) define diffusion-based Boltzmann samplers that approximate a noised energy function from energy evaluations.

The only other flow matching approach using only energy evaluations is Iterated Energy-based Flow Matching (iEFM) (Woo & Ahn, 2024). iEFM adapts the iDEM framework by deriving Monte Carlo estimators for target vector fields. However, it requires significantly more energy evaluations during training and has only been demonstrated on smaller systems (GMM-40, DW-4). Other approaches to training generative models using only energy evaluations include NETS (Albergo & Vanden-Eijnden, 2024), which uses neural transport samplers, methods targeting the reverse diffusive KL divergence as a training objective to mitigate mode-seeking behavior (He et al., 2024), Sequential Controlled Langevin Diffusions (SCLD) (Chen et al., 2025), which connects diffusion models with Sequential Monte Carlo via controlled Langevin dynamics, and Underdamped Diffusion Bridges (Blessing et al., 2025b), which construct bridge processes based on underdamped Langevin SDEs. A common theme among FAB, iDEM, and iEFM is iterative refinement, where models are progressively improved using samples from current model iterations. Our iEWFM algorithm adopts this strategy and, importantly, provides theoretical motivation from a variance-reduction perspective for self-normalized importance sampling, which has previously only been done in Midgley et al. (2022).

**Recent and Concurrent Work.** Several recent and concurrent works have made contributions to energy-based sampling. Approaches leveraging stochastic optimal control include Adjoint Sampling (Havens et al., 2025), the Adjoint Schrödinger Bridge Sampler (Liu et al., 2025), and Trust Region Constrained Measure Transport (Blessing et al., 2025a). Methods us-

Table 3: **Comparison of energy-only methods.** Architecture, training efficiency (energy evaluations), and scalability comparison across state-of-the-art approaches.

| Method | Architecture | Energy Evals | Scalability |
|---|---|---|---|
| FAB (Midgley et al., 2022) | NF | Medium | Limited |
| iDEM (Akhound-Sadegh et al., 2024) | Diffusion | Large | Good |
| **EWFM (This work)** | **CNF** | **Small** | **Good** |

ing annealing strategies include PTSD (Rissanen et al., 2025), PITA (Akhound-Sadegh et al., 2025), and Temperature-Annealed Boltzmann Generators (Schopmans & Friederich, 2025). As these methods appeared recently, we leave systematic comparison for future work.

## 6 CONCLUSION

We introduced iterative Energy-Weighted Flow Matching (iEWFM) and annealed EWFM (aEWFM), novel methods for training continuous normalizing flows as Boltzmann generators without target samples. Our approach combines an energy-reweighted flow matching objective with iterative proposal refinement and temperature annealing for challenging systems. Our evaluation demonstrates competitive sample quality compared to state-of-the-art energy-only methods while requiring up to three orders of magnitude fewer energy evaluations. On high-dimensional systems

(LJ-13, LJ-55), our methods perform comparably or better than the previous state-of-the-art method iDEM, suggesting potential for real-world energy landscapes.

**Limitations.** Several limitations remain. The primary trade-off for our energy evaluation efficiency is the computational cost of CNF density calculations, which dominates wall-clock training time despite our buffering strategy. Additionally, performance gaps on the 8-dimensional DW-4 system compared to FAB and iDEM suggest potential bias in gradient estimates when using the previous model as a proposal distribution, though we lack a full understanding of this behavior.

**Future Work.** Several directions would advance our approach. First, evaluation on larger molecular systems (di-, tetra-, hexapeptides) and systematic comparison with concurrent methods including TA-BG, Progressive Inference-Time Annealing, and Adjoint Sampling (Schopmans & Friederich, 2025; Akhound-Sadegh et al., 2025; Liu et al., 2025; Havens et al., 2025) would be relevant to assess the relative performance of our methods. Second, developing more efficient proposal distributions, such as mixture models, could reduce the computational bottleneck of density evaluation. Third, investigating methodological variations like single-model fine-tuning versus separate model retraining (as done in TA-BG) would provide insights into how one should optimally anneal the temperature.

## REPRODUCIBILITY AND ETHICS STATEMENTS

**Reproducibility.** To support reproducibility, we provide a mathematical derivation of our objective in Appx. B.1, detailed benchmark descriptions and metrics in Appx. C.1 and C.2, implementation details including model architecture and temperature annealing schedule in Appx. C.3, hyperparameter choices in Table 4, and computational environment specifications including GPU types and training times in Appx. C.4. We have provided our implementation and experiments in an anonymous code repository as supplementary material to this submission, enabling reproduction of all reported results.

**Ethics.** We believe there are no significant ethical concerns stemming from our work as it focuses on methodological advances in probabilistic modeling, does not involve human subjects or sensitive data, and poses no immediate societal risks. Nevertheless, we advocate for responsible implementation and use of our methods.

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

## APPENDIX OVERVIEW

This appendix provides additional supporting material for the main text. We organize the content as follows:

- **Appx. A:** A visual illustration of the fundamental Boltzmann sampling problem.
- **Appx. B:** Technical details for the key components of our Energy-Weighted Flow Matching framework, including the complete mathematical derivation of the EWFM objective, the optimal proposal derivation for iEWFM, the complete iEWFM algorithm, details on the amortized training strategy using sample buffers, and weight clipping techniques for stabilizing training.
- **Appx. C:** More detailed specifications of the benchmark systems used in our evaluation, descriptions of the evaluation metrics employed, the full set of implementation details and hyperparameters for reproducibility, and the computational environment used.
- **Appx. D:** Descriptions of the initial Boltzmann generator framework, Flow Annealed Importance Sampling Bootstrap (FAB), and Iterated Denoising Energy Matching (iDEM).
- **Appx. E:** Three potential extensions to improve the efficiency and applicability of our framework: mixture model proposals for more efficient density evaluation, alternative gradient estimation strategies for improved stability, and hybrid approaches that incorporate small amounts of target data when available.
- **Appx. F:** A disclosure of the usage of LLMs during writing and discovery.

## A  THE BOLTZMANN SAMPLING PROBLEM

Fig. 4 visualizes the fundamental challenge of Boltzmann sampling. The left panel shows a two-dimensional energy landscape $E(x)/T$ with two distinct low-energy regions separated by a high-energy barrier. The right panel displays the corresponding target Boltzmann distribution $\mu_{\text{target}}(x) \propto \exp(-E(x)/T)$, where probability mass concentrates precisely in these low-energy regions.

This visualization reveals why traditional trajectory-based methods struggle: to transition between the two modes, a sampling trajectory must cross the high-energy barrier, which occurs rarely. Consequently, methods like MCMC often become trapped in one mode for extended periods, failing to adequately explore the full distribution.

## B  METHODOLOGICAL DETAILS

This section provides technical details for the key components of our Energy-Weighted Flow Matching framework. We begin with the complete mathematical derivation of the EWFM objective, then present the optimal proposal derivation for iEWFM, provide the complete iEWFM algorithm, cover the amortized training strategy using sample buffers, and finally discuss weight clipping techniques for stabilizing training.

### B.1  DETAILED DERIVATION OF THE EWFM OBJECTIVE

In the following, we present the step-by-step mathematical derivation that establishes the theoretical foundation for the EWFM objective introduced in Sec. 3.1.

The EWFM objective addresses the fundamental limitation of conditional flow matching (CFM): its reliance on target samples. The problem is that Boltzmann sampling seeks to generate such samples without access to initial samples. In our strategy we use the fact that Boltzmann distributions have a known unnormalized density $\exp(-E(x)/T)$ and use methods usually applied in importance sampling to transform the CFM loss to not require target samples.

Now let $f(x_1; \theta) = \mathbb{E}_{t \sim U[0,1], X_t \sim p_{t|1}(\cdot|x_1)} \left[ \left\| u_t^\theta(X_t) - u_t(X_t|x_1) \right\|^2 \right]$ denote the expected loss conditioned on an endpoint $x_1$. The CFM loss can then be written as

$$\mathcal{L}_{\text{CFM}}(\theta) = \mathbb{E}_{X_1 \sim \mu_{\text{target}}}[f(X_1; \theta)]. \tag{9}$$

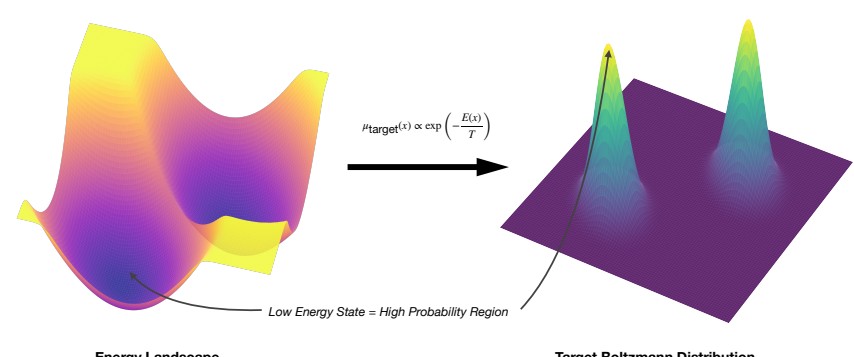

Figure 4: **Illustration of the Boltzmann sampling problem.** *(Left)* A two-dimensional energy landscape $E(x)/T$ with energy values shown in the third dimension, revealing two distinct low-energy regions separated by an energy barrier. *(Right)* The corresponding Boltzmann distribution $\mu_{\text{target}}(x) \propto \exp(-E(x)/T)$, where probability density (shown in the third dimension) is concentrated in low-energy regions. The goal of Boltzmann sampling is to generate samples from this target distribution.

The key insight is that $f(x_1; \theta)$ depends only on the endpoint $x_1$ and not on the distribution from which $x_1$ is sampled. We can therefore use a method usually applied in importance sampling to transform the expectation from the intractable target distribution to an arbitrary proposal distribution $\mu_{\text{prop}}$. Concretely:

$$\mathbb{E}_{X_1 \sim \mu_{\text{target}}}[f(X_1; \theta)] = \int f(x_1; \theta)\, \mu_{\text{target}}(x_1)\, dx_1 \tag{10}$$

$$= \int f(x_1; \theta)\, \frac{\mu_{\text{target}}(x_1)}{\mu_{\text{prop}}(x_1)}\, \mu_{\text{prop}}(x_1)\, dx_1 \tag{11}$$

$$= \mathbb{E}_{X_1 \sim \mu_{\text{prop}}}\left[ \frac{\mu_{\text{target}}(X_1)}{\mu_{\text{prop}}(X_1)} f(X_1; \theta) \right]. \tag{12}$$

For this to be well-defined, we need the support condition $\text{supp}(\mu_{\text{target}}) \subseteq \text{supp}(\mu_{\text{prop}})$.

We now express the density ratio in terms of quantities we can evaluate. Substituting the Boltzmann form $\mu_{\text{target}}(x) = \frac{\exp(-E(x)/T)}{\mathcal{Z}}$ yields

$$\frac{\mu_{\text{target}}(X_1)}{\mu_{\text{prop}}(X_1)} = \frac{\exp(-E(X_1)/T)}{\mathcal{Z}\mu_{\text{prop}}(X_1)} = \frac{w(X_1)}{\mathcal{Z}}, \tag{13}$$

where $w(x) \coloneqq \frac{\exp(-E(x)/T)}{\mu_{\text{prop}}(x)}$. Further, the partition function satisfies $\mathcal{Z} = \int \exp(-E(x)/T)\, dx = \mathbb{E}_{X_1' \sim \mu_{\text{prop}}}[w(X_1')]$ by the same argument as in Eq. (10).

Taken together, this gives us that

$$\mathcal{L}_{\text{CFM}}(\theta) = \mathbb{E}_{X_1 \sim \mu_{\text{target}}}[f(X_1; \theta)] = \mathbb{E}_{X_1 \sim \mu_{\text{prop}}}\left[ \frac{w(X_1)}{\mathbb{E}_{X_1' \sim \mu_{\text{prop}}}[w(X_1')]} f(X_1; \theta) \right] \coloneqq \mathcal{L}_{\text{EWFM}}(\theta; \mu_{\text{prop}}). \tag{14}$$

where $t \sim U[0,1]$, $X_1 \sim \mu_{\text{prop}}$, and $X_t \sim p_{t|1}(\cdot|X_1)$. This establishes the mathematical equivalence $\mathcal{L}_{\text{CFM}}(\theta) = \mathcal{L}_{\text{EWFM}}(\theta; \mu_{\text{prop}})$, ensuring that minimization of our importance-weighted objective yields the same model parameters as the original CFM loss. In practice, EWFM requires only sampling from a proposal $\mu_{\text{prop}}$, evaluating its density, and computing energies $E(x)$, i.e., avoiding the need for target samples entirely.

## B.2 Optimal Proposal Derivation for iEWFM

This section provides the detailed mathematical motivation for the iterative proposal refinement strategy in iEWFM, expanding on the overview presented in Sec. 3.2.

The gradient of the EWFM objective takes the form:

$$\nabla_\theta \mathcal{L}_{\text{EWFM}}(\theta; \mu_{\text{prop}}) = \frac{\mathbb{E}_{X_1 \sim \mu_{\text{prop}}}[\phi_\theta(X_1) w(X_1)]}{\mathbb{E}_{X_1' \sim \mu_{\text{prop}}}[w(X_1')]}, \tag{15}$$

where $\phi_\theta(x_1) = \mathbb{E}_{t \sim U[0,1], X_t \sim p_{t|1}(\cdot|x_1)} \left[ \nabla_\theta \|u_t^\theta(X_t) - u_t(X_t|x_1)\|^2 \right]$ is the gradient of the loss conditioned on $x_1$. If the normalization term in the denominator is estimated from the same samples, which is useful for computational efficiency, the gradient estimator takes the form of a self-normalized importance sampling problem. Given $N$ samples $\{x_1^{(n)}\}_{n=1}^N$ from $\mu_{\text{prop}}$, the corresponding SNIS estimator is

$$\hat{\nabla}_\theta \mathcal{L}_{\text{EWFM}} = \sum_{n=1}^N \tilde{w}^{(n)} \phi_\theta(x_1^{(n)}), \quad \text{where} \quad \tilde{w}^{(n)} = \frac{w(x_1^{(n)})}{\sum_{m=1}^N w(x_1^{(m)})}. \tag{16}$$

From importance sampling theory (Owen, 2013), the optimal proposal distribution that minimizes the variance of the SNIS estimator is

$$\mu_{\text{opt}}(x) \propto \mu_{\text{target}}(x) \cdot \|\phi_\theta(x) - \nabla_\theta \mathcal{L}_{\text{EWFM}}\|. \tag{17}$$

While we cannot sample from $\mu_{\text{opt}}$ directly, as it depends on the gradient we seek, its form suggests an effective approximation strategy. If we make the (relatively strong) simplifying assumption that the difference $\|\phi_\theta(x) - \nabla_\theta \mathcal{L}_{\text{EWFM}}\|$ does not vary substantially across the domain, the optimal proposal becomes approximately the target density, i.e. $\mu_{\text{opt}}(x) \approx \mu_{\text{target}}(x)$. Since our model $q_\theta$ is trained to approximate $\mu_{\text{target}}$, this motivates using $q_\theta$ as the proposal distribution, forming the theoretical motivation for the iterative refinement strategy.

## B.3 Amortized Training with Sample Buffer

A direct implementation of the iterative scheme would be computationally expensive, as evaluating the proposal density $q_\theta(x)$ for each new sample during each gradient step requires solving the reverse-time ODE of the CNF, and computing the corresponding importance weights requires a new energy evaluation. To make this practical, we amortize these costs using a *sample buffer*.

The buffering strategy works as follows: periodically, we generate a fixed set of $N_{\text{buffer}}$ samples and pre-compute both their log-densities under the current proposal $q_\theta$ and their energies $E(x)$. These samples, log-densities, and energy values are cached in a buffer $\mathcal{B} = \{x_j, \log q_\theta(x_j), E(x_j)\}_{j=1}^{N_{\text{buffer}}}$. For multiple subsequent training steps, we then use this static buffer as our proposal distribution by drawing mini-batches from it. By reusing samples multiple times, this buffering approach reduces computational cost and the number of energy evaluations approximately by the average number of times each sample is reused.

The use of a buffer introduces several hyperparameters that represent trade-offs between computational cost and statistical accuracy. The buffer size $N_{\text{buffer}}$ and mini-batch size $N_{\text{batch}}$ control the approximation quality and per-step computational cost, while the buffer refresh rate determines how frequently the buffer is updated to remain consistent with the changing model $q_\theta$. We are currently sampling with replacement from the buffer, but we have also experimented with sampling without replacement, which did not seem to improve performance.

This buffering approach is similar to strategies employed in related work (Midgley et al., 2022; Akhound-Sadegh et al., 2024). The specific hyperparameter values used in our experiments are detailed in Appx. C. You can find the full iEWFM algorithm, including the buffering strategy, in Alg. 2.

## B.4 Stabilizing Training with Weight Clipping

To further mitigate the issue of the SNIS gradient estimator becoming unstable when sampling from distributions with sharp energy landscapes, we investigated weight clipping strategies to cap the influence of samples with very high importance weights.

**Algorithm 2:** Iterative Energy-Weighted Flow Matching (iEWFM) - Detailed

---

**Input:** Energy function $E(x)$; target temperature $T$; initial parameters $\theta$; total epochs $E_{\text{total}}$;
epochs per buffer refresh $E_{\text{refresh}}$; buffer size $N_{\text{buffer}}$; batch size $N_{\text{batch}}$; initial proposal
$\mu_{\text{prop}}^{(0)}$; prior $p_0$.

**Output:** Final trained model parameters $\theta$.

$\mu_{\text{prop}} \leftarrow \mu_{\text{prop}}^{(0)}$;

Generate initial buffer $\mathcal{B}$ by sampling $\{x_j, \log\mu_{\text{prop}}(x_j), E(x_j)\}_{j=1}^{N_{\text{buffer}}}$ from $\mu_{\text{prop}}^{(0)}$;

**for** $e = 1$ **to** $E_{total}$ **do**

    **if** $(e-1) \pmod{E_{refresh}} == 0$ **and** $e > 1$ **then**

        $\mu_{\text{prop}} \leftarrow q_\theta$;

        Generate buffer $\mathcal{B}$ by sampling $\{x_j, \log\mu_{\text{prop}}(x_j), E(x_j)\}_{j=1}^{N_{\text{buffer}}}$ from $\mu_{\text{prop}}$;

    **end**

    Sample mini-batch $\{(x_1^{(n)}, \log\mu_{\text{prop}}(x_1^{(n)}))\}_{n=1}^{N_{\text{batch}}}$ from $\mathcal{B}$;

    Compute importance weights $w^{(n)} = \exp(-E(x_1^{(n)})/T - \log\mu_{\text{prop}}(x_1^{(n)}))$ for
    $n = 1, \ldots, N_{\text{batch}}$;

    **for** $n = 1$ **to** $N_{batch}$ **do**

        Sample $t \sim \mathcal{U}(0,1)$, $x_0 \sim p_0$;

        $x_t \leftarrow (1-t)x_0 + tx_1^{(n)}$;

        $\hat{\phi}_\theta(x_1^{(n)}) \leftarrow \nabla_\theta \left\| u_t^\theta(x_t) - u_t(x_t|x_1^{(n)}) \right\|^2$;

    **end**

    Estimate full gradient $\hat{\nabla}_\theta \mathcal{L}_{\text{EWFM}} = \frac{\sum_{n=1}^{N_{\text{batch}}} \hat{\phi}_\theta(x_1^{(n)}) w^{(n)}}{\sum_{m=1}^{N_{\text{batch}}} w^{(m)}}$;

    Update parameters $\theta \leftarrow \text{OptimizerUpdate}(\theta, \hat{\nabla}_\theta \mathcal{L}_{\text{EWFM}})$;

**end**

**return** $\theta$;

---

We investigated two different clipping strategies to address this issue. The first approach directly clips the negative energy values from above:

$$w_{\text{clipped}}^{(1)}(x) = \frac{\exp\left(\min\left(-E(x)/T, \tau_1\right)\right)}{\log\mu_{\text{prop}}(x)} = \exp\left(\min\left(-E(x)/T, \tau_1\right) - \log\mu_{\text{prop}}(x)\right), \quad (18)$$

while the second approach clips the combined log-importance weight term from above:

$$w_{\text{clipped}}^{(2)}(x) = \exp\left(\min\left(-E(x)/T - \log\mu_{\text{prop}}(x), \tau_2\right)\right), \quad (19)$$

where $\tau_1$ and $\tau_2$ are set to high percentile thresholds (e.g., the 99th percentile) of their respective unclipped terms, preventing the largest importance weights from dominating the gradient estimates. Based on our results during hyperparameter tuning, we adopted the second approach as it performs better, especially when the proposal distribution's likelihood evaluations are less reliable due to the Hutchinson trace estimator used for density evaluation in CNFs.

## C  EXPERIMENTAL DETAILS

To ensure comparability with other recent work, our experimental setup generally follows that of Akhound-Sadegh et al. (2024), employing benchmark systems and metrics commonly used in recent literature on generative Boltzmann sampling. For the performance of the baselines (both iDEM and FAB), we directly cite results from Akhound-Sadegh et al. (2024). We re-implemented an equivalent evaluation pipeline to Akhound-Sadegh et al. (2024) to evaluate our models.

The following subsections provide detailed information on the benchmark systems utilized, the evaluation metrics employed, and the hyperparameters used in our experiments.

## C.1 BENCHMARK SYSTEMS

We evaluate our methods on four classical benchmark systems for Boltzmann generators, covering a range of complexities and dimensionalities. For more detailed descriptions, see Akhound-Sadegh et al. (2024).

**GMM-40** This represents a two-dimensional Gaussian Mixture Model with 40 components arranged on a grid (Midgley et al., 2022). The energy function is the negative log-probability of the mixture: $E(x) = -\log\left(\sum_{i=1}^{40} \frac{1}{40}\mathcal{N}(x|\mu_i, \Sigma_i)\right)$. Despite its low dimensionality, this system challenges models to capture multiple distinct modes.

**DW-4** This system describes four particles in a 2-dimensional space (8D total dimensions) interacting via a double-well potential (Köhler et al., 2020). Following previous work, we set the parameters to $a = 0$, $b = -4$, $c = 0.9$, $d_0 = 4$, and $\tau = 1$. For evaluation, we use ground truth data from Klein et al. (2023).

**LJ-13 and LJ-55** These systems feature clusters of particles governed by the Lennard-Jones potential with harmonic confinement, modeling attractive and repulsive forces between particles. LJ-13 involves 13 particles (39D total), and LJ-55 involves 55 particles (165D total). These systems are particularly challenging due to their high dimensionality and sharp, multi-modal energy landscapes. We use MCMC samples from Klein et al. (2023) as ground truth.

## C.2 EVALUATION METRICS

Sample quality is assessed using two complementary metrics that capture different aspects of distributional similarity:

**2-Wasserstein Distance ($\mathcal{W}_2$)** The $\mathcal{W}_2$ distance quantifies the minimum cost to transform one probability distribution into another:

$$\mathcal{W}_2(\mu, \nu) = \left(\inf_{\pi \in \Pi(\mu,\nu)} \int_{\mathbb{R}^d \times \mathbb{R}^d} \|x - y\|_2^2 \, d\pi(x, y)\right)^{1/2} \tag{20}$$

where $\Pi(\mu, \nu)$ is the set of all joint distributions with marginals $\mu$ and $\nu$. We estimate this distance by computing the Wasserstein distance between the empirical distributions of generated and ground truth samples using the Python Optimal Transport package (Flamary et al., 2021) with Euclidean distance. Lower $\mathcal{W}_2$ values indicate a closer distributional match.

**Negative Log-Likelihood (NLL)** The NLL measures how likely a set of ground truth test samples is under the learned generative model. Following the evaluation pipeline from Akhound-Sadegh et al. (2024), we first generate a large dataset of samples from our trained model (100,000 for GMM-40, DW-4, and LJ-13; 10,000 for LJ-55). We then train a separate evaluation CNF on these generated samples. The NLL is then the negative log-probability of the ground truth test data under this evaluation CNF, computed using the Instantaneous Change of Variables Formula. For the GMM-40, DW-4, and LJ-13 tasks, we use exact computation of the divergence term, while for the high-dimensional LJ-55 system, we use the Hutchinson trace estimator. Lower NLL values indicate better sample quality.

While we employ this evaluation pipeline for comparability with Akhound-Sadegh et al. (2024), it has limitations: the optimal checkpoint for the evaluation CNF is selected using a validation set of samples from the target distribution, and we also found that there were slight discrepancies between NLL values under our original model versus the evaluation model. We note that we do not find this evaluation procedure optimal and plan to use different metrics that do not require training a second model in future versions of this paper.

**Omission of Effective Sample Size (ESS)** We omit the Effective Sample Size metric used in Akhound-Sadegh et al. (2024) as we found it to be relatively unstable during our evaluations. Specifically, the ESS values varied significantly depending on whether we used the density from our original model or the density from the evaluation CNF model, and also showed considerable variation

between different runs. Additionally, we note that Akhound-Sadegh et al. (2024) evaluated the ESS on only 16 test samples for all tasks, which may not provide reliable estimates.

**Energy Function Evaluations**   As energy evaluations can be computationally expensive for large systems (Klein & Noé, 2024; Havens et al., 2025), we report the total number of energy function evaluations during training as an important efficiency metric.

### C.3   Implementation Details and Hyperparameters

**Baselines**   We compare our algorithms against two state-of-the-art methods for energy-only Boltzmann sampling: Flow Annealed Importance Sampling Bootstrap (FAB) (Midgley et al., 2022), which combines normalizing flows with Annealed Importance Sampling, and iterated Denoising Energy Matching (iDEM) (Akhound-Sadegh et al., 2024), which employs a score-matching approach. Both methods enable training Boltzmann generators without target data. For both baselines, we report performance metrics from Akhound-Sadegh et al. (2024).

In addition to our main proposed methods (iEWFM and aEWFM), we also evaluate a simplified variant we call EWFM, which represents an ablation of iEWFM without the iterative refinement component. Instead of using the current model as a proposal, EWFM employs a simple, fixed proposal distribution throughout training (e.g., a standard Gaussian). This allows us to assess the specific contribution of the iterative scheme to the overall performance.

**Model Architectures**   To ensure direct comparability with baseline results, we use the same network architectures as Akhound-Sadegh et al. (2024) used for score estimation to parameterize our vector fields: a multi-layer perceptron (MLP) with sinusoidal positional embeddings for the GMM task, and an Equivariant Graph Neural Network (EGNN) (Satorras et al., 2021) for the DW-4, LJ-13, and LJ-55 systems. All models were optimized using the Adam optimizer (Kingma & Ba, 2014).

**Training Details**   We employ weight clipping on the top percentiles of importance weights to stabilize training, with clipping percentiles ranging from 97.5% to 99.9%. We found that the choice of clipping percentile was often one of the most impactful hyperparameters for training stability and final performance. When training our models with either iEWFM or aEWFM, we evaluate model densities using the Hutchinson trace estimator for the divergence calculations, which leads to imperfect density estimates but is required to keep training computationally tractable. For evaluation, we use the final checkpoint at the end of training without any model selection based on validation performance.

For aEWFM, we employ a geometric temperature annealing schedule starting from $T_{\text{init}} = 10.0$ and decreasing every 2 epochs until reaching the target temperature of 1.0, using a total of 100 epochs to anneal across all systems. The buffer refresh rate is set to once per epoch across all experiments, though for the more challenging LJ systems, we reduced the refresh rate and increased the number of mini-batches per epoch to save computational cost. Hyperparameters were chosen through a structured search over key parameters, including learning rates, buffer sizes, and clipping percentiles.

### C.4   Computational Environment

Experiments were conducted on a variety of NVIDIA GPUs. The computationally intensive LJ-55 experiments utilized an H100 GPU. The LJ-13 experiments were performed on an A100 GPU, while the smaller GMM-40 and DW-4 systems were trained on a GTX 1080 Ti GPU.

For reference, the training times were as follows: LJ-55 (H100) took 25-27 hours, LJ-13 (A100) took 30-31 hours, DW-4 (GTX 1080 Ti) took 10 hours, and GMM-40 (GTX 1080 Ti) took 5 hours. We note that both wall-clock time and energy evaluation counts could likely be improved further by reducing the number of training epochs, as convergence was often achieved in approximately one-half of the total training time, with only marginal improvements observed in later stages.

Table 4: **Key hyperparameters across the various benchmark systems.** The main hyperparameters (learning rate through mini-batches per epoch) are used identically for EWFM, iEWFM, and aEWFM. The annealing schedule parameters are only used for aEWFM, while iEWFM and EWFM use a fixed temperature of 1.0 throughout training. For DW-4, the clipping percentile notation 97.5%/99.9% indicates that 97.5% was used for aEWFM and iEWFM while 99.9% was used for EWFM.

| Hyperparameter | GMM-40 | DW-4 | LJ-13 | LJ-55 |
|---|---|---|---|---|
| Learning Rate | $5 \times 10^{-4}$ | $1 \times 10^{-3}$ | $5 \times 10^{-4}$ | $5 \times 10^{-4}$ |
| Buffer Size ($N_{\text{buffer}}$) | 5000 | 5000 | 5000 | 500 |
| Batch Size ($N_{\text{batch}}$) | 5000 | 5000 | 5000 | 500 |
| Total Training Epochs | 5000 | 2500 | 2500 | 2500 |
| Mini-batches per Epoch | 10 | 10 | 20 | 20 |
| *Annealing Schedule (for aEWFM)* | | | | |
| Initial Temperature ($T_{\text{init}}$) | 10.0 | 10.0 | 10.0 | 10.0 |
| Epochs per Temperature | 2 | 2 | 2 | 2 |
| Total Annealing Epochs | 100 | 100 | 100 | 100 |
| *Weight Clipping* | | | | |
| Clipping Percentile | 99.9% | 97.5%/99.9% | 99.9% | 98.0% |

# D   FURTHER DETAILS ON RELATED WORK

This section provides descriptions of the initial Boltzmann generator framework, Flow Annealed Importance Sampling Bootstrap (FAB), and Iterated Denoising Energy Matching (iDEM).

## D.1   THE INITIAL BOLTZMANN GENERATOR METHOD

The core contribution of Boltzmann generators, as introduced by Noé et al. (2019), is to adapt the training of generative models to the setting where only the energy function $E(x)$ and temperature $T$ are known. The initial training objective is to minimize the reverse KL divergence, $\text{KL}(q_\theta \| \mu_{\text{target}})$, which, unlike the forward KL, can be estimated using samples from the model $q_\theta$ itself:

$$
\begin{aligned}
\text{KL}(q_\theta \| \mu_{\text{target}}) &= \int q_\theta(x) \log \frac{q_\theta(x)}{\mu_{\text{target}}(x)} dx \\
&= \mathbb{E}_{X \sim q_\theta} \left[ \log q_\theta(X) + E(X)/T \right] + \log \mathcal{Z}.
\end{aligned}
\tag{21}
$$

Since $\log \mathcal{Z}$ is constant with respect to the model parameters $\theta$, it can be ignored during optimization. Note that the reverse KL divergence has the known limitation of being "mode-seeking", meaning that optimizing it can lead to the model collapsing to only a subset of the modes of a multi-modal target distribution. To stabilize training, Noé et al. (2019) use a combination of the forward and reverse KL divergences:

$$
\lambda \text{KL}(\mu_{\text{target}} \| q_\theta) + (1 - \lambda) \text{KL}(q_\theta \| \mu_{\text{target}})
\tag{22}
$$

where $\lambda$ is a hyperparameter, which is often annealed during training. Here, the forward KL term is estimated using a small set of training samples from the target distribution.

## D.2   FLOW ANNEALED IMPORTANCE SAMPLING BOOTSTRAP (FAB)

Flow Annealed Importance Sampling Bootstrap (FAB) (Midgley et al., 2022; 2023) augments a normalizing flow $q_\theta$ with an Annealed Importance Sampling (AIS) bootstrap to minimize the mass-covering $\alpha$-divergence (with $\alpha = 2$) between the target $\mu_{\text{target}}$ and the flow:

$$
\mathcal{D}_2\big(\mu_{\text{target}} \| q_\theta\big) \;=\; \tfrac{1}{2} \int_{\mathbb{R}^d} \frac{\mu_{\text{target}}(x)^2}{q_\theta(x)} \, dx,
\tag{23}
$$

whose minimizer yields the lowest possible variance of importance weights. This choice of divergence addresses the mode-seeking behavior of the reverse KL divergence used in earlier methods.

During training, AIS is run with the current flow as the initial distribution and a target density $g(x) \propto \mu_{\text{target}}(x)^2 / q_\theta(x)$; this makes the AIS path focus on regions that contribute most to $\mathcal{D}_2$, similar to how our iterative approach uses the current model as the proposal for the next iteration to minimize the variance of the estimator. Each AIS run returns pairs $\{(x, w_{\text{AIS}})\}_{i=1}^N$, used to update the flow through a self-normalized surrogate loss:

$$\mathcal{S}'(\theta) = -\mathbb{E}_{\text{AIS}}\big[\bar{w}_{\text{AIS}} \log q_\theta(x)\big], \qquad \bar{w}_{\text{AIS}} = \frac{w_{\text{AIS}}}{\sum_{i=1}^N w_{\text{AIS},i}}. \tag{24}$$

Furthermore, a prioritized replay buffer lets FAB reuse past AIS samples, reducing computational costs. FAB requires significantly fewer energy evaluations than previous methods, though it has limited scalability to more complex systems.

## D.3 Iterated Denoising Energy Matching (iDEM)

Iterated Denoising Energy Matching (iDEM) (Akhound-Sadegh et al., 2024) trains a diffusion-based sampler by replacing standard score-matching with a Denoising Energy Matching objective. This is done by constructing a Monte-Carlo estimator of the score for noised points $x_t \sim \mathcal{N}(x_1, \sigma_t^2 \mathbf{I})$:

$$\hat{s}_K(x_t, t) = \frac{\frac{1}{K}\sum_{i=1}^K \exp\left(-\mathcal{E}\left(x_{1|t}^{(i)}\right)\right) \nabla \exp\left(-\mathcal{E}\left(x_{1|t}^{(i)}\right)\right)}{\frac{1}{K}\sum_{j=1}^K \exp\left(-\mathcal{E}\left(x_{1|t}^{(j)}\right)\right)}, \quad x_{1|t}^{(1)}, \dots, x_{1|t}^{(K)} \sim \mathcal{N}\left(x_t, \sigma_t^2\right),$$
$$\tag{25}$$

and fits a score network via the loss $\mathcal{L}_{\text{DEM}}(\theta) = \mathbb{E}_{t,x_t}[\|s_\theta(x_t, t) - \hat{s}_K(x_t, t)\|^2]$. Training proceeds in two coupled loops with a replay buffer strategy. Similar to our approach, iDEM uses an iterative refinement strategy where the model is progressively improved by using samples from the current model to train the next iteration. The method requires gradients of the energy function and, while it scales well to high-dimensional systems, requires relatively many energy evaluations. Notably, iDEM was the first method to successfully train using only energy evaluations on the challenging 55-particle Lennard-Jones system.

# E Further Extensions

In addition to the core algorithms presented in the main text, we explored several extensions that could further improve the efficiency and applicability of our framework. This section details three potential directions: mixture model proposals for computational efficiency, alternative gradient estimation strategies for improved stability, and hybrid approaches that incorporate small amounts of target data when available.

## E.1 Mixture Model Proposals for Improved Efficiency

While using the current model $q_\theta$ as the proposal distribution in iEWFM is well-motivated, it is computationally expensive, and the likelihood evaluations can be inaccurate, potentially leading to unreliable importance weights.

To address these limitations, we explored approximating the proposal distribution more efficiently using buffer samples from the current model $q_\theta$. One approach is to use kernel density estimation (KDE) to approximate the current model distribution based on the buffer samples. Given buffer samples $\{x_i\}_{i=1}^N$ from the current model $q_\theta$, we construct the smoothed proposal distribution as

$$\mu_{\text{prop}}(x) = \frac{1}{N}\sum_{i=1}^N K_h(x - x_i), \tag{26}$$

where $K_h$ is a kernel function with bandwidth $h$. For instance, using a Gaussian kernel yields a Gaussian mixture model as the proposal distribution. This "smoothed" proposal distribution $\mu_{\text{prop}}$ provides efficient sampling and density evaluations.

However, our experiments revealed that the quality we achieve with this approach depends strongly on the kernel bandwidth $h$, which controls the smoothness of the KDE approximation. We found

an interesting trade-off: smaller bandwidths improved the Wasserstein distance but worsened the negative log-likelihood performance, suggesting potential bias in the learned distribution. Due to these concerns, we ultimately did not adopt this approach for our main experiments, though we believe it represents an interesting direction for future work.

### E.2 ALTERNATIVE GRADIENT ESTIMATION STRATEGY

An alternative strategy for estimating the EWFM objective, which we did not implement in this work, involves rewriting it as a set of nested expectations. This approach may improve the stability of the estimation, particularly for target distributions with sharp modes. The objective can be expressed as

$$\mathcal{L}_{\text{EWFM}} = \mathbb{E}_{t \sim U[0,1], X_t \sim p_t} \left[ \mathbb{E}_{X_1 \sim p_{1|t}(\cdot|X_t)} \left[ \frac{w(X_1)}{\mathbb{E}_{X_1' \sim \mu_{\text{prop}}}[w(X_1')]} \left\| u_t^\theta(X_t) - u_t(X_t|X_1) \right\|^2 \right] \right], \quad (27)$$

where the outer expectation is over the marginal path $p_t$ and the inner is over the posterior path $p_{1|t}$. One could then generate the required samples as follows: first draw $X_t$ by interpolating between a prior sample and a proposal sample, then draw $X_1$ for the inner expectation by sampling from the posterior path conditioned on $X_t$. However, this formulation is incompatible with our efficient sample buffer strategy, as the inner expectation requires repeated sampling of new $X_1$ samples conditioned on $X_t$, which prevents pre-computation of the proposal log-densities and energy evaluations.

### E.3 COMBINING WITH TARGET SAMPLES

While our methods are designed for settings without target samples, a potential extension could be to leverage a small dataset if one exists. This might help stabilize the initial stages of training, particularly for the iEWFM algorithm. Following the original Boltzmann generator framework (Noé et al., 2019), one could explore a hybrid loss function

$$\mathcal{L}_{\text{hybrid}}(\theta; \lambda) = (1 - \lambda)\mathcal{L}_{\text{EWFM}}(\theta; \mu_{\text{prop}}) + \lambda\mathcal{L}_{\text{CFM}}(\theta), \quad (28)$$

where $\lambda \in [0, 1]$ is a weighting parameter. The $\mathcal{L}_{\text{EWFM}}$ component would still benefit from the iterative proposal scheme. To mitigate overfitting to the small dataset, it may be beneficial to anneal the hyperparameter $\lambda$ during training, starting with a higher value and gradually decreasing it to zero.

It is important to note that this hybrid approach cannot be directly combined with our aEWFM algorithm. The available target samples are from the distribution at the final target temperature $T$, not the higher intermediate temperatures $T_i > T$ used in the annealing schedule. Integrating target samples into aEWFM would require a more complex procedure, such as using importance resampling to adapt the target samples to different temperatures.

## F LLM USAGE DISCLOSURE

During the preparation of this manuscript, large language models (LLMs) were used in a supporting role. For writing, we employed LLMs to provide grammar and style suggestions; no proofs or derivations were generated by LLMs. In addition, we used advanced reasoning models to challenge our own interpretations and conclusions.

For retrieval and discovery, we employed LLM-based Deep Research tools to help identify potentially relevant references and to occasionally summarize external work for scoping purposes.

Overall, LLMs served as an auxiliary tool for writing polish, exploratory literature discovery, and internal cross-checking, without contributing novel technical content.

