# OpenReview forum: "Energy-Weighted Flow Matching: Unlocking Continuous Normalizing Flows for Efficient and Scalable Boltzmann Sampling"
_ICLR.cc/2026/Conference — Submitted to ICLR 2026_

### Official Review · Reviewer_oi2V · 2025-10-18

**Soundness:** 2
**Presentation:** 2
**Contribution:** 1
**Rating:** 2
**Confidence:** 5

**Summary:**

This paper proposes an iterative method for training neural samplers. The core idea is to use importance weighting to refine the standard flow-matching training objective. On top of this reweighting scheme, the authors introduce a temperature-annealing schedule that trains the sampler progressively—starting from a high-temperature distribution and gradually moving toward the target temperature.

While the paper is clearly written, I have several concerns. First, the related-work coverage omits a number of recent and highly relevant papers. For neural samplers, [1] improves upon iDEM via a bootstrapping technique and reports markedly stronger results on LJ55 than iDEM. There are also competitive neural samplers that should be cited and compared against, such as [2, 3]. Regarding annealing to aid neural-sampler training, several works should be cited as well, including [4, 5, 6] (acknowledging that [6] is very recent).

Second, the methodological novelty appears limited. Importance-weighted regression objectives have been explored previously—for example, [7] applies an analogous idea to (denoising) score matching. Finally, relying solely on importance reweighting raises scalability concerns in higher dimensions, since importance weights typically become more concentrated (“weight degeneracy”) as dimensionality grows.

In sum, despite the clear exposition, the paper lacks coverage of related work and the proposed method offers limited novelty. I therefore lean toward rejection.

[1] OuYang, RuiKang, Bo Qiang, and José Miguel Hernández-Lobato. "Bnem: A boltzmann sampler based on bootstrapped noised energy matching." arXiv preprint arXiv:2409.09787 (2024).

[2] Chen, Junhua, et al. "Sequential controlled langevin diffusions." arXiv preprint arXiv:2412.07081 (2024).

[3] Blessing, Denis, et al. "Underdamped diffusion bridges with applications to sampling." arXiv preprint arXiv:2503.01006 (2025).

[4] Rissanen, Severi, et al. "Progressive Tempering Sampler with Diffusion." arXiv preprint arXiv:2506.05231 (2025).

[5] Akhound-Sadegh, Tara, et al. "Progressive Inference-Time Annealing of Diffusion Models for Sampling from Boltzmann Densities." arXiv preprint arXiv:2506.16471 (2025).

[6] Blessing, Denis, et al. "Trust Region Constrained Measure Transport in Path Space for Stochastic Optimal Control and Inference." arXiv preprint arXiv:2508.12511 (2025).

[7] Yu, Ziyang, Wenbing Huang, and Yang Liu. "Force-guided bridge matching for full-atom time-coarsened dynamics of peptides." arXiv preprint arXiv:2408.15126 (2024).

**Strengths:**

1. The paper is clearly written and easy to follow.

2. Empirically, the proposed method outperforms FAB and iDEM while requiring fewer energy evaluations.

**Weaknesses:**

1.	Related work coverage is incomplete. For training neural samplers, [1] improves upon iDEM via a bootstrapping technique and shows substantially better performance on LJ55 than iDEM. Additional strong neural samplers, such as [2, 3], should be cited and compared against. For annealing-assisted training, relevant works include [4, 5, 6] (with [6] being very recent).

2.	Limited methodological novelty. Using importance sampling to reweight a regression-style training loss is not new; closely related ideas appear in prior work, e.g., importance-weighted (denoising) score matching [7]. The paper should clarify what is fundamentally new relative to these approaches.

3.	Empirical concerns and scalability:

- On the 2D GMM task, EWFM outperforms its iterative and annealed variants (iEWFM, aEWFM), which is surprising in a low-dimensional setting where buffer/annealing should not degrade performance.

- On LJ55, there is a visible gap between EWFM samples and ground truth, as reflected in both the interatomic-distance and energy histograms. This raises concerns about scalability of the loss reweighting: importance weighting is known to suffer from the curse of dimensionality (weight degeneracy).

- Moreover, the 2D GMM with a prior covering most modes (as in iDEM) is relatively simple; [2] demonstrates scalability to a 50D case, setting a higher bar for comparison.

4.	Ablation: loss reweighting is underexplored. The same importance weights could be used to train a standard diffusion/score-matching model by replacing the conditional flow-matching loss \|u_\theta - u\|^2 with a conditional denoising score-matching loss \|s_\theta - s\|^2. An ablation comparing these two paths would clarify whether the benefit comes from the reweighting itself or from the specific flow-matching formulation.

5.	Ablation: annealing is underexplored. The proposed annealing scheme could also be applied to iDEM by tempering the energy used in the Monte-Carlo score estimator. An ablation demonstrating this would help isolate the contribution of annealing versus the rest of the method.

[1] OuYang, RuiKang, Bo Qiang, and José Miguel Hernández-Lobato. "Bnem: A boltzmann sampler based on bootstrapped noised energy matching." arXiv preprint arXiv:2409.09787 (2024).

[2] Chen, Junhua, et al. "Sequential controlled langevin diffusions." arXiv preprint arXiv:2412.07081 (2024).

[3] Blessing, Denis, et al. "Underdamped diffusion bridges with applications to sampling." arXiv preprint arXiv:2503.01006 (2025).

[4] Rissanen, Severi, et al. "Progressive Tempering Sampler with Diffusion." arXiv preprint arXiv:2506.05231 (2025).

[5] Akhound-Sadegh, Tara, et al. "Progressive Inference-Time Annealing of Diffusion Models for Sampling from Boltzmann Densities." arXiv preprint arXiv:2506.16471 (2025).

[6] Blessing, Denis, et al. "Trust Region Constrained Measure Transport in Path Space for Stochastic Optimal Control and Inference." arXiv preprint arXiv:2508.12511 (2025).

[7] Yu, Ziyang, Wenbing Huang, and Yang Liu. "Force-guided bridge matching for full-atom time-coarsened dynamics of peptides." arXiv preprint arXiv:2408.15126 (2024).

**Questions:**

1. 2D GMM: Why does EWFM perform better than its iterative and annealing variants (iEWFM, aEWFM) on the 2D GMM task? What failure modes did you observe for the iterative buffer or the annealing schedule?

2. LJ13/LJ55 W2 computation: When reporting the (data) Wasserstein-2 distances on LJ13 and LJ55, did you account for translation and rotation (i.e., SE(3) equivariance) in the metric, e.g., via alignment or quotienting out rigid motions?

3. Metrics reported: Why report only data W2 and NLL? Could you also include energy-W2, MMD, and total-variation distance to provide a more complete picture of sample quality and energy consistency?

---

> ### Author Response · Authors · 2025-11-25
>
> We thank the reviewer for their comments and valuable feedback on our paper. Below, we address the concerns and questions raised.
>
> 1. **Additional related work**
>
> We thank the reviewer for making us aware of these works. We have cited these in the updated version of the paper. Regarding comparison against \[1-3\], we were unfortunately unable to benchmark against these during the rebuttal due to computational and time constraints.
>
> 2. **Clarification on why EWFM performs better on the 2D-GMM**
>
> The better performance of the non-iterative EWFM on the 2D GMM is not surprising. In standard EWFM, we use a fixed (Gaussian) proposal distribution where the density $q(x)$ is known exactly. In iEWFM, the proposal is a CNF where the density is approximated. In low dimensions, the exact Gaussian covers the modes sufficiently well, and the noise from the CNF density estimation in iEWFM outweighs the benefit of proposal adaptation. This trade-off flips in high dimensions, where the iterative refinement of iEWFM/aEWFM becomes necessary.
>
> 3. **Concerns regarding the Scalability of EWFM and its variants**
>
> We agree that importance weighting suffers from degeneracy in high dimensions. This is exactly the reason why we need a good proposal distribution (via iterative refinement) to keep the weights well-behaved. While there is a gap on LJ-55, we emphasize that we perform competitively with the iDEM, which was until relatively recently SOTA on this task, while using significantly fewer energy evaluations.
>
> 4. **Ablations: Annealing**
>
> We agree that the proposed experiment would be an interesting experiment to isolate the contribution of annealing; we unfortunately did not have the time or computational resources to implement and tune this modification to the baseline method during the rebuttal.
>
> 5. **Accounting for translation and rotation for Wasserstein-2 distance**
>
> For the results reported in Table 1 (where some numbers slightly changed due to re-runs of experiments), we did not account for translation and rotation in the metric; we simply used the Euclidean distance to maintain consistency with previous papers. We agree that this is a good point for future evaluation.
>
> 6. **Reporting further metrics**
>
> That is a fair point. We agree that Energy-$W\_2$ and MMD would provide a more complete picture. However, generating these metrics was not possible due to computational and time constraints during the rebuttal.

---

> > ### Comment · Reviewer_oi2V · 2025-11-26
> > **Response to author's rebuttal**
> >
> > The reviewer thanks the authors for their efforts on introducing related works that are lack in the first edition of the manuscript and clarifying the reviewer's concerns. However, the main concerns of the proposed work are not solved:
> >
> > 1. On the scalability of the methodology itself:
> >  - SNIS is not scalable in high dimensional spaces, especially when the target density only appears in the importance weights.
> >
> > 2. On the scalability of the proposed method in practice, and its novelty:
> >  - Works like **[7]** in the reviewer's initial review, have explored using IS to reweigh flow matching or denoising score matching losses. While the author's work doesn't principally improve the scalability of this method, nor showcase the practical stability in high dimensional toy tasks such as high-dimensional GMM (say, 50D in **[2]** in the reviewer's initial review) and LJ55. That is, neither the novelty and the practicallity of the proposed method are shown.
> >
> > And also some minor issues:
> >  -  considering the rotational equivariance when calculating the metrics are important, as it increases the variance of metrics which can introduce bias in the analysis, **if** the rotational differences of generated samples are uniformly (wrt the O(3) group) random. If not, imagine that one model constantly outputs samples with a rotation R wrt to the ground-truth samples used for evaluation, and the other model doesn't, this introduces very high bias and unfair comparison.
> >
> >  - Regarding to SOTA in LJ55, the reviewer believes both [1] in the reviewer's initial review and Adjoint Sampling (https://arxiv.org/abs/2504.11713) demonstrate better performance on this task. While Adjoint Sampling (https://arxiv.org/abs/2504.11713) is even more scalable to more complex tasks.
> >
> > Overall, as a consequence of the above main issues 1&2, the reviewer would not change the score.

---

### Official Review · Reviewer_pxHr · 2025-11-01

**Soundness:** 2
**Presentation:** 2
**Contribution:** 1
**Rating:** 2
**Confidence:** 3

**Summary:**

The authors propose a framework for sampling from Boltzmann distributions using the conditional flow matching method. Unlike typical flow matching methods which require access to samples from the target density, the authors leverage importance sampling to leverage a different proposal distribution. Since the ground-truth energy is known and computable up to a normalizing constant, the importance weight can be easily computed; allowing training of flow-matching models through self-normalized importance sampling. Empirical evaluation of the proposed method shows that it is competitive with iterated Denoising Energy Matching while requiring significantly fewer energy evaluations.

**Strengths:**

- The authors extend the flow-matching framework to sampling problems where the target density is available up to a normalization constant, but samples from it are not available. This is an important direction of research towards AI for science applications.
- The proposed method is simple to understand and is built on top of well established methods: flow matching and self-normalized importance sampling.
- Empirical evaluation shows that the proposed method outperforms baselines on high-dimensional problems as well as enjoys fewer energy evaluations.

**Weaknesses:**

- The baselines considered in the work are rather weak. The comparisons are only made with FAB and iDEM but the authors should be comparing their proposed method against [1-8].
- The novelty of the work is severely limited. It is well known that iDEM, iEFM, etc. are based on self-normalized importance sampling objectives, hence such systems are already using SNIS approaches to learn either a score or drift. In this context, what is the proposed method adding since this is already well known (eg. see TweeDEM in [2]).
- The work does not provide any quantitative evidence of training and inference times (in wall clock), which is a very important metric since the authors highlight that their method is more efficient.

**References**

[1] Woo, Dongyeop, and Sungsoo Ahn. "Iterated energy-based flow matching for sampling from boltzmann densities." arXiv preprint arXiv:2408.16249 (2024).

[2] OuYang, RuiKang, Bo Qiang, and José Miguel Hernández-Lobato. "Bnem: A boltzmann sampler based on bootstrapped noised energy matching." arXiv preprint arXiv:2409.09787 (2024).

[3] Havens, Aaron, et al. "Adjoint sampling: Highly scalable diffusion samplers via adjoint matching." arXiv preprint arXiv:2504.11713 (2025).

[4] Choi, Jaemoo, et al. "Non-equilibrium Annealed Adjoint Sampler." arXiv preprint arXiv:2506.18165 (2025).

[5] Albergo, Michael S., and Eric Vanden-Eijnden. "Nets: A non-equilibrium transport sampler." arXiv preprint arXiv:2410.02711 (2024).

[6] Chen, Junhua, et al. "Sequential controlled langevin diffusions." arXiv preprint arXiv:2412.07081 (2024).

[7] Vargas, Francisco, et al. "Transport meets variational inference: Controlled monte carlo diffusions." arXiv preprint arXiv:2307.01050 (2023).

[8] Sendera, Marcin, et al. "Improved off-policy training of diffusion samplers." Advances in Neural Information Processing Systems 37 (2024): 81016-81045.

**Questions:**

- It is quite unclear to me how the proposed method reduces the number of energy evaluations needed. In fact, does the proposed method reduce to IEFM?

---

> ### Author Response · Authors · 2025-11-25
>
> We thank the reviewer for their comments and valuable feedback on our paper. Below, we address the concerns and questions raised.
>
> 1. **Additional comparisons to more methods**
>
> We agree that it would be interesting to benchmark against the suggested methods \[1-8\]. However, due to computational and time constraints, we are not able to provide these during the rebuttal period. The choice of iDEM and FAB was made since, to the best of our knowledge at the time, these were the two most commonly used and strongest methods used earlier this year (before multiple recent and partially concurrent papers came out).
>
> 2. **Clarification of the differences to iDEM and iEFM**
>
> We respectfully disagree with the comment that our work lacks novelty or reduces to iEFM. The difference is that iDEM and iEFM use a self-normalized importance objective to construct a Monte Carlo estimator of the vector field (or score) at every timestep. We do not estimate the score; instead, we simply re-weight the individual flow matching losses against the exact conditional vector field. This bypasses the noisy score-estimation step entirely. Importantly, this means that our method does not reduce to iEFM. We have added a remark in subsection 3.1 to clarify this.
>
> 3. **Reduction of Energy Evaluations**
>
> The reduction in energy evaluations comes from the difference in how EWFM and its variants utilize energy information. iDEM uses a nested estimation scheme where, for every training step, it constructs a Monte Carlo estimator of the score at new noisy points $x\_t$. This requires fresh, expensive evaluations of energy gradients $\\nabla E(x\_t)$ at new noisy points $x\_t$ that cannot be effectively cached. In contrast, our importance weights $w(x\_1)$ only depend on the clean proposal samples $x\_1$. This decoupling allows us to compute $E(x\_1)$ once, cache the values in the buffer, and reuse them across multiple gradient updates. This difference allows us to extract significantly more training signal per energy evaluation.
>
> 4. **Wall-clock times**
>
> We do provide our wall-clock training times in Appendix C.4. While we are not more efficient than iDEM in terms of wall-clock time on these simpler problems (due to CNF integration), EWFM and its variants provide an alternative training method that is advantageous on problems where energy evaluations are more expensive/the bottleneck.

---

> > ### Comment · Reviewer_pxHr · 2025-11-27
> >
> > Thanks for the clarification. I am curious to know why it is not more efficient thatn iDEM during training? Reduction in energy evaluations should make the training faster, no? I also fail to understand where the CNF integration plays a role in training?
> >
> > Additionally, while all my theoretical concerns have been addressed (except for above), I am hesitant to increase my score primarily because the work lacks a detailed comparison with a lot of related work; without this comparison it is really hard to evaluate the contribution of the proposed method.

---

> > > ### Author Response · Authors · 2025-11-28
> > >
> > > Thanks for the response and the further questions.
> > >
> > > For the weights $w\left(x_1\right)=\frac{\exp \left(-E\left(x_1\right) / T\right)}{\mu_{\text {prop }}\left(x_1\right)}$ in the Energy-Weighted Flow-Matching Objective, we need to evaluate $\mu_{\text {prop }}\left(x_1\right)$, i.e., the density of data points under the proposal distribution. When the proposal distribution is the current generative model $q_\theta$ as it is in iEWFM and aEWFM, we then need to evaluate the density under the current generative model, which leads to a CNF integration being required.
> > >
> > > These CNF integrations are relatively expensive, which is the primary reason why our methods have relatively long wall-clock times.

---

### Official Review · Reviewer_Pfo1 · 2025-11-03

**Soundness:** 2
**Presentation:** 2
**Contribution:** 2
**Rating:** 2
**Confidence:** 5

**Summary:**

This paper introduces an alternative yet equivalent formulation of the flow matching loss that removes the need for target samples when training CNFs to generate samples from the Boltzmann distribution; instead, the approach adopts an expectation over an arbitrary proposal distribution combined with importance sampling to recover the true target distribution. While the method is conceptually appealing, its practical scalability remains up for debate. The largest system evaluated in the paper is LJ55, which is considerably smaller than those addressed by state-of-the-art approaches for Boltzmann sampling. Comparisons are drawn to iDEM, which has since been surpassed by newer samplers such as adjoint sampling (arXiv preprint arXiv:2504.11713). Other recent developments, including discrete normalizing flow methods that scale to much larger systems (arXiv preprint arXiv:2508.18175; arXiv preprint arXiv:2502.18462), further highlight the gap in demonstrating scalability (granted these use data and not energy alone). Continuous normalizing flow models such as ECNF++ have also been successfully applied to larger systems (arXiv preprint arXiv:2502.18462) demonstrating scalable performance. Overall, while the proposed approach is theoretically interesting, the experimental validation and benchmarking fall short of convincingly establishing its effectiveness at scale.

**Strengths:**

The proposed mathematical framework is conceptually appealing, with its primary benefit demonstrated through a reduction in the number of energy evaluations relative to comparable approaches. This efficiency may offer practical advantages for Boltzmann sampling applications.

**Weaknesses:**

The benchmarking presented in the paper is very limited. Other standard benchmarks like PIS or LogVariance have not been contrasted on the same datasets. Annealing is introduced, yet benchmarking relative to the most performant annealing models (PITA: arXiv preprint arXiv:2506.16471) is not performed. Further, the proposed approach shows only marginal improvements over iDEM, while several more recent methods (e.g., arXiv preprint arXiv:2504.11713) have demonstrated superior performance. These newer approaches are not included in the comparison and are instead deferred to future work; however, without such evaluations, the practical utility and scalability of the proposed method remain uncertain.

**Questions:**

- How do the number of energy evaluations compare with adjoint sampling?
- Can ESS and energy-W1/W2 be included as additional metrics for performance?
- How does the performance on ESS and energy-W1/W2 contrast relative to adjoint sampling?
- How does EWFM scale to simple peptides e.g., alanine dipeptide, tri-alanine, etc.
- Can more powerful and expressive networks be used for training EWFM, and how do they perform?
- On the annealing experiments, how does the approach perform relative to PITA (arXiv preprint arXiv:2506.16471)?

---

> ### Author Response · Authors · 2025-11-25
>
> We thank the reviewer for their comments and valuable feedback on our paper. Below, we address the concerns and questions raised.
>
> 1. **Benchmarking on more benchmarks (PIS, LogVariance) and against new approaches**
>
> We agree that benchmarking on more datasets (PIS, LogVariance) and against newer approaches such as PITA and Adjoint Sampling would be interesting. Due to computational and time constraints, conducting these additional large-scale experiments was unfortunately not feasible during the rebuttal period. We would furthermore like to mention that PITA is, in our understanding, considered concurrent work under the ICLR guidelines (published less than four months before the submission deadline).
>
> 2. **Comparison Of Number of Energy Evaluations Against Adjoint Sampling**
>
> On the benchmarking problems provided by us, Adjoint Sampling does need roughly one to two orders of magnitude fewer energy evaluations than our method. At the same time, we note that Adjoint Sampling requires the gradient of the energy function $\\nabla E(x)$. Computing gradients typically requires backpropagation through the energy function, which can be computationally expensive. In contrast, EWFM and its variants only need the energy function values $E(x)$, making them applicable even in settings where differentiable energy functions are not available or prohibitively expensive.
>
> 3. **Additional metrics, larger scale benchmarks and more expressive networks**
>
> We agree these would be valuable additions. However, setting up experiments for peptide systems or training more expressive networks was not feasible within the time of the rebuttal period.
>
> 4. **Comparison to PITA**
>
> Given that PITA has been reported to outperform iDEM – which performs comparably to our method – we expect that PITA would likely outperform our method in terms of the quality of the generated samples as well. Additionally, PITA uses a similar magnitude of energy evaluations to EWFM and its variants on the benchmarks we have in common.
>
> However, we note that PITA also requires the gradient of the energy function for its "Target Score Matching" objective during training, which imposes the same computational and differentiability constraints mentioned above compared to our gradient-free approach.

---

### Official Review · Reviewer_Yp6F · 2025-11-04

**Soundness:** 2
**Presentation:** 2
**Contribution:** 3
**Rating:** 4
**Confidence:** 5

**Summary:**

This paper introduces Energy-Weighted Flow Matching (EWFM), a novel and theoretically elegant data-free objective for training Continuous Normalizing Flows (CNFs) to sample from unnormalized Boltzmann distributions. The core idea, which reformulates Conditional Flow Matching via importance sampling, is a solid and principled contribution. Based on this, the authors propose an iterative (iEWFM) and an annealed (aEWFM) algorithm.

**Strengths:**

The paper's primary contributions are clear and theoretically sound:
1. The EWFM objective is a novel and principled method for training CNFs on unnormalized distributions without target data. The use of importance sampling to adapt the CFM loss is an elegant and effective idea.
2. The authors build on this objective to propose two practical algorithms: iterative EWFM (iEWFM), which progressively refines its own proposal distribution, and annealed EWFM (aEWFM), which incorporates a temperature schedule to handle more complex energy landscapes.
3. The method's main strength is its efficiency in terms of energy function calls.

**Weaknesses:**

Despite the novel objective, the empirical validation of the method suffers from several critical flaws that undermine the paper's claims of state-of-the-art performance and efficiency:
1. The most significant drawback is the complete failure to compare against Iterated Energy-based Flow Matching (iEFM). The authors explicitly identify iEFM as the "only other flow matching approach using only energy evaluations". This omission is highly concerning, especially since iEFM was successfully demonstrated on the GMM-40 and DW-4 benchmarks, the ones where this paper's method shows significant limitations.
2. The claims of "competitive" performance are not fully supported by the results. The method fails catastrophically on DW-4 system, producing a W2 distance much worse than the baselines. It also underperforms against both iDEM and FAB on the LJ-13 system. The authors' brief hypothesis of "potential bias" for the DW-4 failure is insufficient and does not inspire confidence in the method's robustness.
3. The method for reporting statistical significance in Table 1 is non-standard. The paper bolds results based on the mean $\pm$ 1 SD interval. This is not a rigorous test of statistical significance and can be misleading, as it may incorrectly group methods with very different mean performances. A standard Welch's t-test would be required for a valid comparison.

**Questions:**

Your review correctly identifies the relevance of Bayesian inference tasks, such as those in [1]. For these problems (e.g., Bayesian logistic regression on the German Credit  or Breast Cancer  datasets), the "energy function" is the negative log-posterior, and its evaluation requires a full pass over the dataset. Given that your method's primary bottleneck is the 30+ hour wall-clock time for CNF density calculations , how do you justify its "efficiency" for this common class of problems, where the cost of an energy evaluation and the cost of a model density evaluation may be comparable?


References:

[1] Denis Blessing, Xiaogang Jia, Johannes Esslinger, Francisco Vargas, and Gerhard Neumann. Beyond ELBOs: A large-scale evaluation of variational methods for sampling

---

> ### Author Response · Authors · 2025-11-25
>
> We thank the reviewer for their comments and valuable feedback on our paper. Below, we give our answers regarding the expressed concerns and state the changes we have made in response.
>
> 1. **Comparison against Iterated Energy-based Flow Matching (iEFM)**
>
> We agree that a comparison against iEFM would be interesting. Since iEFM only provided results on GMM-40 and DW-4, and since to the best of our knowledge the code is not available online, evaluating it on the larger-scale benchmarks would have required a full re-implementation and extensive hyperparameter tuning from scratch. Unfortunately, this was not feasible during the rebuttal period.
>
> 2. **Competitive Performance Not Fully Supported?**
>
> We agree that the fact that we do not have a good explanation for our suboptimal performance on DW-4 is a limitation of our work. However, we respectfully disagree that our performance is "catastrophic." We are comparing against methods that were close to SOTA or were SOTA on these benchmarks earlier this year. Furthermore, we would argue that our method does perform competitively with FAB and iDEM on the other systems. For example, while we do underperform FAB and iDEM on LJ-13 in terms of NLL, we reach comparable or slightly better $W\_2$ scores on this system (cf. Table 1).
>
> 3. **Bold values via Welch’s t-test**
>
> We agree on this point with the reviewer and have updated Table 1 accordingly to use Welch’s t-test. Please note that we also took this opportunity to re-run a few of our experiments to ensure robustness, which led to slight updates in the reported values.
>
> 4. **Efficiency in Bayesian Inference Tasks**
>
> We are not sure whether we understand the question raised here correctly. Is the question: How can we justify the ‘efficiency’ of our method for use cases where the cost of energy evaluations is large?
>
> If this is the question, we would argue that this is precisely the case where our method has the largest advantage compared to other methods. In settings where the energy function (e.g., the negative log-posterior on a large dataset) is significantly more expensive than the CNF density evaluation, our method is very efficient because it needs significantly fewer energy evaluations than other methods while still reaching a similar performance (cf. Table 2).

---

### Meta-Review · Area_Chair_R1W7 · 2025-12-28

**Summary:**

All reviewers agreed that the proposed method is elegant and theoretically sound. They highlighted that the low number of energy function calls is a plus (Yp6F, pxHr). However, the reviewers criticize that a significant body of related work is missing. While many of these works are recent and could probably be considered parallel to the present work at this point, the authors agreed that a comparison to several baselines would be interesting. Reviewers also criticize the high overall wall clock costs, partially mitigating the reduction in energy evaluations. The authors also did not provide additional requested metrics on their trained models and samples although this would probably have been feasible during the rebuttal.

Given their number, the concerns still outweigh the positive technical contribution. I am convinced that a resubmission that closely argues the choice of baselines and that emphasizes and materializes the advantages of the method (only scalar energy evaluations, good caching) would have significantly larger impact than the current paper. I therefore propose to reject. I strongly encourage the authors to adopt a more self-confident communication in the rebuttal. I think that focusing the rebuttal on the advantages of the method, and a sharper distinction to prior and clearly labeled concurrent work might have turned the decision around.

**Reviewer Concerns:**

The main criticism of all reviewers is that several baselines are missing. While many of the additional baselines were actually concurrent work under ICLR work, several baselines remain uncompared: Underdamped Diffusion Bridges (Blessing et al. ICLR 2025), Vargas et al. ICLR 2024, Sendera et al. NeurIPS 2024, Woo et al. arXiv 2024 (iEFM; I could not find that this work is published anywhere, but the reviewers and authors took this very seriously, so a comparison appears adequate), Albergo et al. ICLR 2025.

Also, additional requested metrics were not reported.

Regarding scaling [oi2V]: From a random sample of concurrent papers, scaling up to LJ-55 seems common in the field. So I do not consider this a point against the paper.

**Reviewer Scores:**

- Yp6F: Score probably would have increased given the statistically sound evaluation in Table 1, presumably clearing the discussion about which methods is best on the reported experiments.
- Pfo1: Score probably would not have been increased, as additional metrics were not reported by the authors even though they had the chance to post again. A proper argument on when the gradient of the energy is significantly more expensive than the value could have turned the discussion.
- pxHr: Presumably unchanged, since the major concern is the lack of baselines.
- oi2V:  The reviewer explicitly stated they are concerned about scaling and the related work, after a discussion with the authors. The authors did not provide further arguments.

---

### Decision · Program_Chairs · 2026-01-26

Reject